# Conductive and elastic bottlebrush elastomers for ultrasoft electronics

Pengfei Xu [1], Shaojia Wang[1], Angela Lin [2], Hyun-Kee Min[3,4,5], Zhanfeng Zhou[1], Wenkun Dou [1], Yu Sun [1], Xi Huang [3,4,5], Helen Tran [2,6] ✉ & Xinyu Liu [1,7] ✉

Understanding biological systems and mimicking their functions require electronic tools that can interact with biological tissues with matched softness. These tools involve biointerfacing materials that should concurrently match the softness of biological tissue and exhibit suitable electrical conductivities for recording and reading bioelectronic signals. However, commonly employed intrinsically soft and stretchable materials usually contain solvents that limit stability for long-term use or possess low electronic conductivity. To date, an ultrasoft (i.e., Young's modulus <30 kPa), conductive, and solvent-free elastomer does not exist. Additionally, integrating such ultrasoft and conductive materials into electronic devices is poorly explored. This article reports a solvent-free, ultrasoft and conductive PDMS bottlebrush elastomer (BBE) composite with single-wall carbon nanotubes (SWCNTs) as conductive fillers. The conductive SWCNT/BBE with a filler concentration of 0.4 − 0.6 wt% reveals an ultralow Young's modulus (<11 kPa) and satisfactory conductivity (>2 S/m) as well as adhesion property. Furthermore, we fabricate ultrasoft electronics based on laser cutting and 3D printing of conductive and non-conductive BBEs and demonstrate their potential applications in wearable sensing, soft robotics, and electrophysiological recording.

Soft and stretchable electronics can enhance existing capabilities in mimicking sensing behaviors of biological systems and measuring vital signs reflecting general physiological states, enabling biomimetic designs to improve interactions with humans[1–3]. In particular, there is an increasing demand in ultrasoft (i.e., Young's modulus, $E < 30$ kPa[4]) electronics for applications ranging from deep-sea applications of ultra-gentle robotic actuators[5] to artificial robotic skins[6] and human-machine interfaces[2] (Fig. 1a). A key materials challenge to address for these applications is replicating the dynamic mechanical characteristics (e.g., softness and stretchability) of biological systems while still maintaining high performance (e.g., electronic conductivity and mechanical durability) for long-term use. There is an inherent

mechanical mismatch between biological tissues that have a Young's modulus ranging from 10 Pa to 1 MPa[4,7] (Fig. 1b) and conventional inorganic electronics (i.e., rigid Utah array[8]) that have a Young's modulus in the range of 1–200 GPa[9,10], leading to device failure over time and hindering long-term commercial utility. Engineering approaches on designing flexible and stretchable device architectures such as incorporating serpentine[11–13] and/or mesh structures[14,15] have improved elasticity of electronics[16] and can accommodate certain levels of strain, but their limited stretchability and inherently higher modulus can result in unintended immunological responses and/or local damage to tissue[17,18]. Another approach is to use intrinsically stretchable materials (e.g., hydrogels[19–22] and ionogels[23,24]); these

[1]Department of Mechanical and Industrial Engineering, University of Toronto, Toronto, Ontario M5S 3G8, Canada. [2]Department of Chemistry, University of Toronto, Toronto, Ontario M5S 3H6, Canada. [3]Program in Developmental and Stem Cell Biology, The Hospital for Sick Children, Toronto, Ontario M5G 1X8, Canada. [4]Arthur and Sonia Labatt Brain Tumour Research Centre, The Hospital for Sick Children, Toronto, Ontario M5G 1X8, Canada. [5]Department of Molecular Genetics, University of Toronto, Toronto, Ontario M5S 3E1, Canada. [6]Department of Chemical Engineering and Applied Chemistry, University of Toronto, Toronto, Ontario M5S 3E5, Canada. [7]Institute of Biomedical Engineering, University of Toronto, Toronto, Ontario M5S 3G9, Canada. ✉e-mail: tran@utoronto.ca; xyliu@mie.utoronto.ca

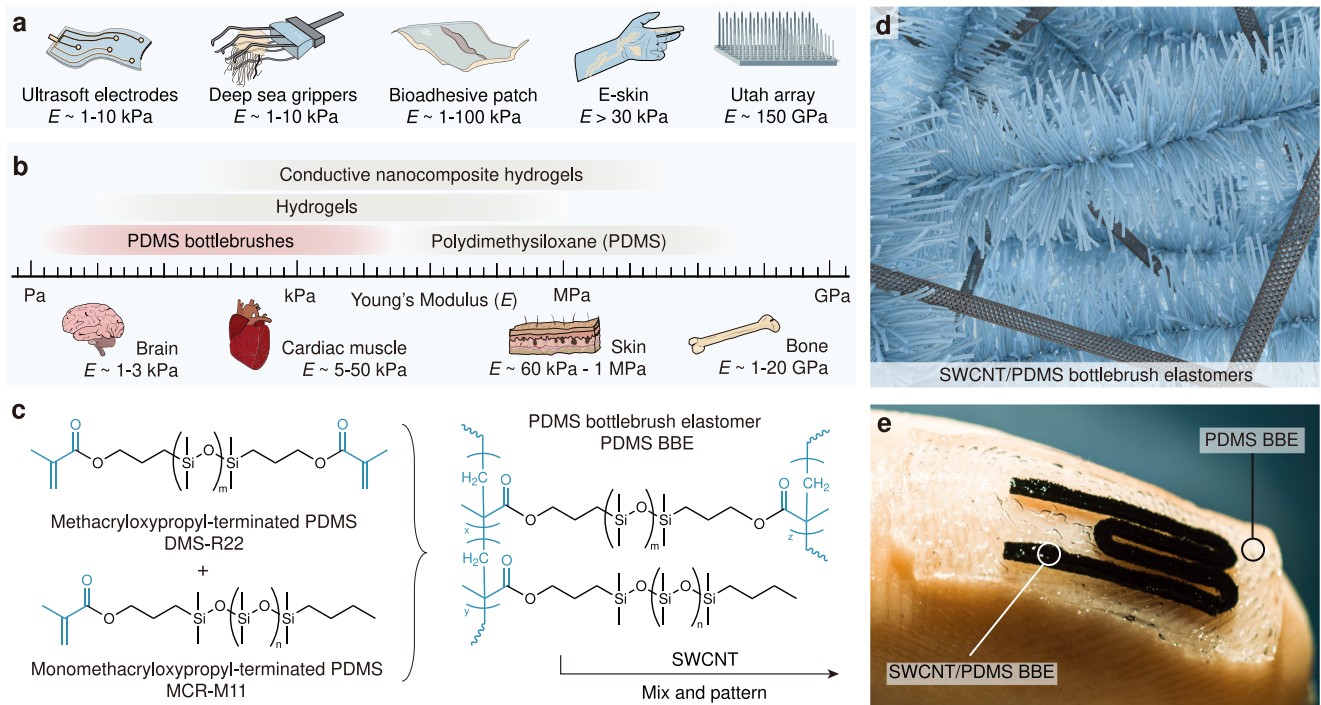

**Fig. 1 | Design of ultrasoft and conductive BBE composites. a** Different applications require soft and/or stretchable materials with different levels of Young's modulus ($E$). In comparison, conventional electronic materials (silicon and metals) applied on rigid electronics (e.g., the Utah array) have a much higher $E$ than intrinsically stretchable materials. **b** The approximate Young's modulus range of different biological tissues and commonly used intrinsically stretchable materials. Hydrogels and BBEs are two main materials that can match the softness of tissues. **c** Chemical structures of PDMS crosslinkers DMS-R22, monomers MCR-M11, and crosslinked BBEs. **d** A schematic showing the nanometer-scale composition of conductive PDMS BBE containing bottlebrushes and SWCNTs (dimensions are not to scale). **e** An integrated ultrasoft electronic made of the conductive SWCNT/BBE and non-conductive pure PDMS BBE.

materials contain high water content or expensive ionic liquids that can diffuse, which limits the environments in which they can be employed. Ultrathin polymer sheets or nanomeshes are nice examples that can effectively reduce mechanical constraints and increase conformability to arbitrary surfaces[25–27], but the ultrathin structures with in-plane patterns require complex device fabrication process. Alternatively, bottlebrush elastomers (BBEs) are a class of intrinsically stretchable materials that can achieve ultra-low Young's modulus without solvents, because of their highly branched architecture consisting of polymeric side chains attached to a polymer backbone, leading to reduced entanglements in comparison to linear analogs[28–33]. For example, polydimethylsiloxane (PDMS)-based BBEs can achieve a Young's modulus of <1 kPa[28,34], comparatively lower than that of commercial PDMS linear elastomers such as the Sylgard 184 with a Young's modulus of 100 kPa to 3 MPa[35,36] (Fig. 1b). Moreover, the PDMS BBE can maintain the similar biocompatibility and chemical resistance of its linear counterpart as the chemical composition remains the same. However, existing BBE have primarily been composed of non-conductive polymers (e.g., PDMS), limiting their application in soft and stretchable electronics[28,37,38].

One straightforward approach to render BBEs conductive is to incorporate conductive fillers (e.g., carbon nanotubes, metallic nanowires and flakes) and form conductive BBE composites. Material design considerations on the potential toxicity of such conductive fillers and the polarity difference between the matrix elastomers and the fillers need to be carefully addressed[39–41] (Supplementary Fig. 1). To this end, Self et al.[42] recently reported the first conductive BBE based on poly(4-methylcaprolactone) mixed with carbon nanotube (CNT) fillers, which provides a relatively low modulus of 66 kPa and an improved electrical conductivity of 0.01 S/m. The BBE preparation in this work involves a multi-step chemical synthesis of monomers and crosslinkers, which is a

sophisticated process for non-experts. The modulus of the conductive BBE of 66 kPa is slightly higher than the typical modulus range of the soft tissues (0–30 kPa), and there is opportunity to improve the electrical conductivity (0.01 S/m) to construct high-performance electronics.

Here we report the preparation of a solvent-free conductive PDMS BBE with single-walled carbon nanotubes (SWCNTs) as the conductive filler (Fig. 1c–e), and explore the device fabrication methods and applications based on the SWCNT/BBE. The SWCNT/BBE reveals a Young's modulus range of 2.98–10.65 kPa and a conductivity level of 2.06–17.84 S/m, five times softer than that of the only previously reported conductive BBE[42]. To the best of our knowledge, this is also the softest conductive solvent-free elastomers ever reported. To simplify the material synthesis and popularize its practical use in ultrasoft electronics, commercially available PDMS monomers, crosslinkers, and SWCNTs were employed to prepare the conductive BBE (Fig. 1c), and no complicated chemical synthesis was involved. To fabricate ultrasoft electronic devices, we developed laser cutting and three-dimensional (3D) printing-based processes for layer-by-layer patterning of conductive and non-conductive BBEs into ultrasoft strain sensors, touch pads, and wearable electrodes. To highlight the advantages of using BBE-based ultrasoft electronics in different applications, we demonstrate use of these devices as ultrasoft strain sensors on rigid- and soft-bodied robots and worms, and as wearable electrodes for on-body electrocardiogram (ECG) recording. This work reports the first patterning of conductive BBEs via 3D printing and laser cutting to construct fully BBE-based electronic devices. With the increasing demand of an ultrasoft and conductive material platform to be employed in robotic and biomedical applications, our SWCNT/BBE provides a solution to build ultrasoft electronics, and will potentially expand our capabilities of understanding biological systems and mimicking their functions.

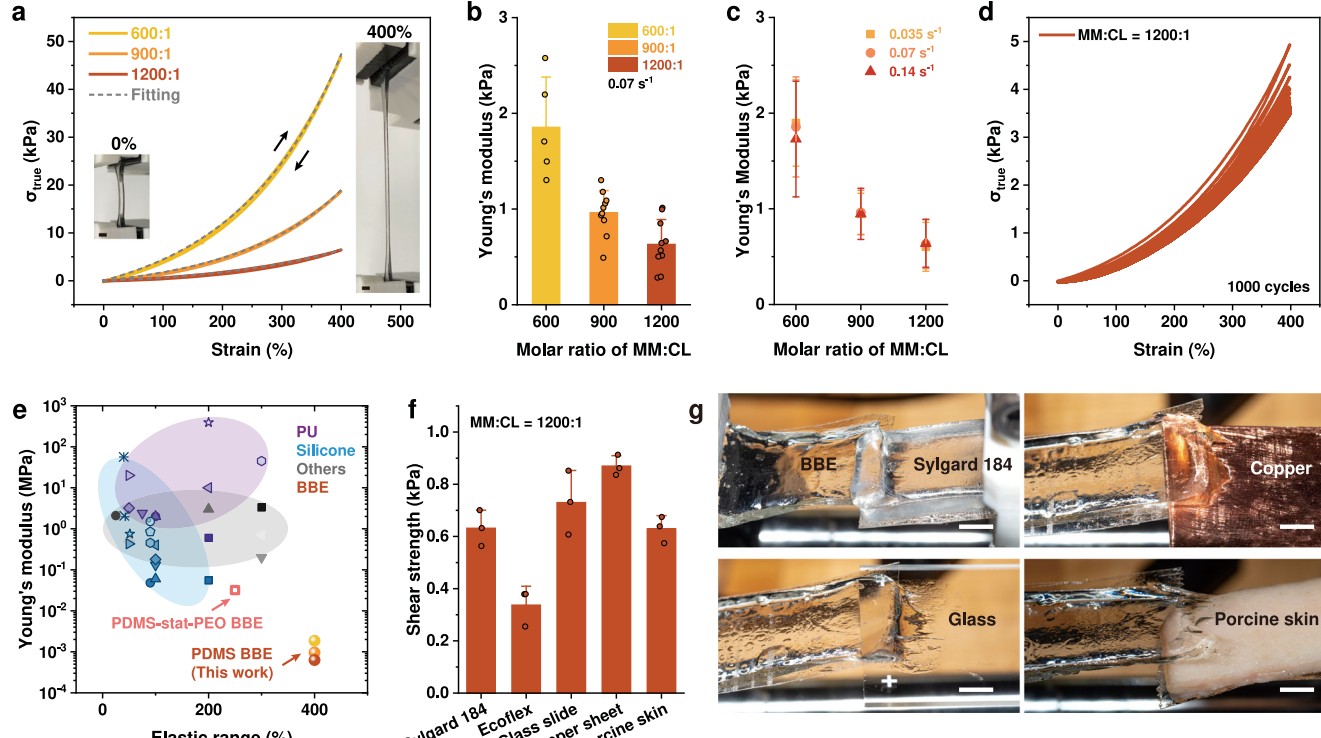

**Fig. 2 | Characterizations of pure PDMS BBEs. a** Cycling tests of PDMS BBEs with different crosslinking ratios (molar ratio of MM:CL = 600:1, 900:1, and 1200:1) at the strain of 400%. The dash line represents the fitting curve obtained from the stress-strain model for unentangled polymer networks. The inset shows photographs of BBE under stretching. **b** The Young's modulus of PDMS BBEs with different crosslinking ratios. **c** The comparison of Young's modulus of PDMS BBEs obtained from the cyclic tests under different strain rates (0.035 s⁻¹, 0.070 s⁻¹, and 0.14 s⁻¹). **d** Long term cycling tests (1000 cycles at the strain of 400%) of the PDMS BBE with the crosslinking ratio (MM:CL) of 1200:1. **e** The Ashby-style plot of elastic range and Young's modulus of different elastomers and PDMS BBE in this work. **f** The adhesive shear strength between the PDMS BBE (with the crosslinking ratio (MM:CL) of 1200:1) and different substrates, including PDMS Sylgard 184, Ecoflex 00-10, glass slide, copper sheet, and porcine skin. **g** Photographs of the adhesion test between the PDMS BBE and different substrates. The scale bar is 5 mm. Error bars denote the standard deviation of the measurements.

## Results

### Ultrasoft PDMS BBEs: preparation and characterization

To enable the ultra-softness of BBEs, the polymer architecture plays a vital role. By adjusting synthesis parameters such as the length of polymer side chains, the distance between side chains (grafting density), and the distance between crosslinks (network strand length), one can synthesize BBEs replicating the stress−strain behaviors of different kinds of biological tissues[28]. Longer side chains and higher grafting density reduce the physical entanglements among polymers, and larger network strand length leads to fewer crosslinks, thus lower Young's modulus and higher stretchability of the material. Although the intricate tailoring of bottlebrush polymers enables new morphology[32,43], mechanical properties[44], and crosslinking strategies[45–47] for existing BBEs (Supplementary Table 1), complicated procedures, low overall yield and reproducibility of the existing synthesis methods could hinder practical applications of ultrasoft elastomers. To this end, Vatankhah-Varnosfaderani[28] has proposed to use commercially available PDMS monomers to polymerize bottlebrush polymer networks by a "grafting-through" approach. Hu et al.[38] recently reported a one-reaction method to simply prepare super-soft silicone elastomers. While compared to grafting side chains onto a polymeric backbone, the "grafting-through" method allows us to form bottlebrush polymers with higher grafting density[48], and potentially provide a lower Young's modulus. Thus, in this work, we selected commercially available PDMS monomers (MCR-M11) and crosslinkers (DMS-R22) to prepare BBEs via one-pot free-radical polymerization[34], a simple method that can be readily adopted by non-chemists (Fig. 1c). We selected azobisisobutyronitrile (AIBN) as the initiator to crosslink the BBE thermally (Fig. 1c)

as the SWCNT fillers to be added will make the material opaque and thus not amenable to the common photocrosslinking methods[28,44]. Additionally, no solvents were used in the synthesis to ensure high stability of the prepared BBE for long-term use.

With the commercially available monomers MCR-M11 and crosslinkers DMS-R22, the length of side chains and grafting density are fixed, leaving the third parameter−network strand length−available to be modulated to tune the softness of BBEs. Here, the crosslinking ratio (molar ratio of monomer:crosslinker (MM:CL)) was adjusted at three levels of 600:1, 900:1, and 1200:1 to modulate the network strand length. Shorter strands generally reduce the extension of polymer networks, resulting in an increase of $E$ (0.63 kPa to 1.85 kPa) with the increase of crosslinker weight percentage (wt%) (Fig. 2a, b). The model-fitting results of the unentangled networks[28] show a good agreement with our experimental results (Supplementary Fig. 2 and Supplementary Table 2), indicating limited polymer entanglements in elastomers that induce the ultralow $E$. Due to the covalently crosslinked polymer strands and reduced polymer entanglements, we hypothesize that the PDMS BBE can keep a high elasticity in a wide strain range, which is crucial for the stability requirement of matrix materials for ultrasoft electronics. With cyclic tensile testing at 50%, 100%, 200%, and 400% strains at a fixed strain rate of 0.070 s⁻¹ (rate of deformation: 50 mm/min) (Fig. 2a, Supplementary Fig. 2), all BBEs at the three crosslinking ratios display a fully reversible deformation (Supplementary Fig. 3) and only small changes in the Young's modulus (Supplementary Fig. 4), and hysteresis within 20 cycles (Supplementary Fig. 5). Moreover, the overlayed stress−strain curves and the consistent Young's modulus of elastomers at different strain rates (0.035 s⁻¹, 0.070 s⁻¹, and 0.14 s⁻¹)

indicate that the elasticity is independent of strain rates (Fig. 2c, Supplementary Figs. 6a-n). The dynamic sweeps further indicate the elastic regime (0.01 rad/s to 1 rad/s) of the BBE [Supplementary Fig. 6o]. Compared to previous linear elastomers and BBEs, our PDMS BBE possesses high elasticity in a wide strain range as well as an ultralow Young's modulus (Fig. 2e, Supplementary Table 3). In order to maintain a low Young's modulus for CNT-filled BBEs, the lowest crosslinking level (crosslinking ratio of MM:CL = 1200:1) was chosen to synthesize the non-conductive elastomer matrix because of its ultra-low Young's modulus (average: 0.63 kPa) and high elasticity (elastic range of 400% strain) over 1000 loading-unloading cycles (Fig. 2d, Supplementary Fig. 7).

In addition to the ultra-softness and high elasticity, we observed that the pure PDMS BBE inherently possesses satisfactory interfacial adhesion with different material surfaces. Lap shear testing of the BBE (crosslinking ratio: 1200:1) was conducted on surfaces of Sylgard 184 elastomer, Ecoflex, glass slide, copper sheet, and porcine skin (Fig. 2g and Supplementary Fig. 8), and the average shear strength was measured to be 0.63 kPa, 0.34 kPa, 0.73 kPa, 0.87 kPa, and 0.63 kPa, respectively (Fig. 2f). Although this level of shear strength is lower than those (>10 kPa) of commonly used adhesive materials[49], it is sufficient for securely attaching a piece of BBE to different surfaces in dry conditions, which can withstand its own gravity and different strain levels during operation, and even lift objects such as a ping-pong ball or a beaker (potentially useful as a pick-place mechanism; see Supplementary Fig. 9 and Supplementary Movie 1). From the lap shear testing (Fig. 2g and Supplementary Fig. 8), one can see that the BBE left a material trace propagating along the adhesion interface, leading to

significant deformation of the BBE. Different from ionic or covalent bonds of common adhesive materials that lead to high shear strength through the "lock and key" mechanism[50], the adhesive property of hydrophobic PDMS BBE is possibly derived from the Van der Waals force[51,52]. This could be attributed to the highly-branched long side chains and ultra-softness of the solvent-free BBE that maximize contact surfaces as well as conformity when interfacing with different substrates[34].

## Conductive SWCNT/BBEs: preparation and characterization

Previous study has shown the difficulty of fitting the Young's modulus of conductive materials into the range of soft tissues due to the use of highly entangled linear elastomer. Thus, in this work, the tissue-matched low Young's modulus is our first priority when designing conductive BBEs. Although the SWCNT can enhance the conductivity, a balance point should be selected where the conductive BBEs have a low Young's modulus as well as a satisfactory conductivity. To prepare conductive BBEs, we first mixed SWCNTs with the precursor of pure PDMS BBE and then cured the precursor to form a SWCNT percolation network in the elastomer matrix. The similar polarity of the SWCNTs[41] and the BBE allowed them to be easily mixed and cured into a conductive composite (see Methods section). The high aspect ratio (length:width of ~2500) of the SWCNTs rendered good conductivity of the SWCNT/BBEs with a relatively low filler-loading concentration (0.2–0.6 wt%), which also leads to a favorable stretchability (>100%) and post-aging elasticity of the material (Fig. 3a and Supplementary Fig. 10). The scanning electron microscopy (SEM) images of the SWCNT/BBEs with different loading concentrations (0.2 wt%, 0.4 wt%,

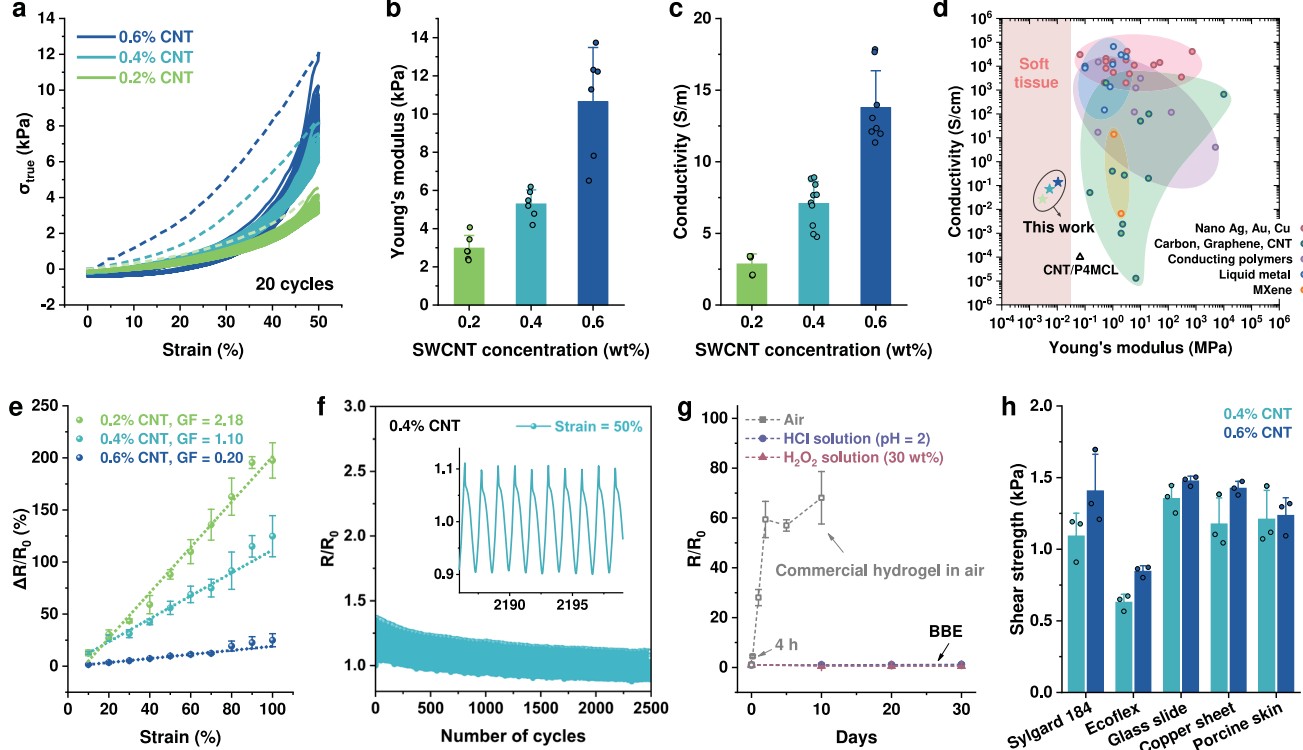

**Fig. 3 | Characterization of the conductive SWCNT/BBE. a** Cycling tests of PDMS BBEs with 0.2 wt%, 0.4 wt%, and 0.6 wt% SWCNT at the strain of 50%. 20 cycles were conducted, with the first half cycle represented by the dash line. **b** The Young's modulus and **c** conductivity of PDMS BBEs with 0.2 wt%, 0.4 wt%, and 0.6 wt% SWCNT. **d** The Ashby-style plot of the conductivity and Young's modulus of different conductive elastomers with metallic nano materials, carbon-based materials, conducting polymers, liquid metal, or MXene as conductive components. Hydrogels are not included as they contain water and are not solvent-free. Data points are

from references labeled in Supplementary Fig. 12 and Supplementary Table 4. **e** Normalized change in resistance as a function of tensile strain. **f** Cyclic durability (2500 cycles) of the normalized change in resistance for BBE with 0.4 wt% CNT under cyclic loading to 100% strain. The inset shows a representative resistance change within the cyclic measurements. **g** Environment stability of the normalized change in resistance for SWCNT/BBE stored in HCl, $H_2O_2$ solutions or exposed to air. **h** The adhesive shear strength between SWCNT/BBE and different substrates. Error bars denote the standard deviation of the measurements.

and 0.6 wt%) show the distributions of SWCNTs on the elastomer surfaces where the SWCNT density increased with the loading concentration. While the cross-sectional SEM images show that there are SWCNT bundles in the elastomer matrix (Supplementary Fig. 11), possibly derived from the pristine entangled SWCNTs. At SWCNT concentrations of 0.2 wt%, 0.4 wt%, and 0.6 wt%, the Young's modulus and the electrical conductivity of the SWCNT/BBE were measured to be 2.98 kPa, 5.29 kPa, and 10.65 kPa (Fig. 3b) and 2.68 S/m, 7.08 S/m, and 13.78 S/m (Fig. 3c), respectively. A higher concentration (>0.6%) of SWCNT would make the Young's modulus higher (>15 kPa), and also make the SWCNT more difficult to be dispersed in the BBE precursor during the preparation process. Thus, we selected 0.6 wt% as the highest concentration in this work. Compared to the ultralow Young's modulus (0.63 kPa) of the pure PDMS BBE, the addition of the SWCNT fillers into the elastomer matrix increased the material's Young's modulus because of the reinforcement effect of the SWCNT network; however, even the highest Young's modulus (10.65 kPa) of our SWCNT/BBE is much lower than those of conductive dry elastomers and also five times lower than that of the only previously reported conductive BBE[42] (Fig. 3d, Supplementary Fig. 12 and Supplementary Table 4). This could be attributed to our different chemical components of the bottlebrushes and looser crosslinking networks. And the modulus range of 2.98–10.65 kPa well fits within the range of soft biological tissues. The electrochemical impedance spectroscopy characterization reveals the frequency-independent impedance and zero phase angle of SWCNT/BBE (Supplementary Fig. 13), indicating the pure electron-based conductivity stemming from SWCNT. The conductivity of our SWCNT/BBE is lower than those of metallic nanomaterials based conductors, possibly due to the higher contact resistance and lower inherent conductivity of SWCNT[53]. However, we have succeeded to fit the Young's modulus of our conductive SWCNT/BBE into the range of soft tissues, and with the satisfactory conductivity, the SWCNT/BBE is believed to have more potentials than pure BBE and commonly used elastomers.

By virtue of its ultra-softness, good stretchability and conductivity, our SWCNT/BBE is a promising material candidate for ultrasoft and stretchable sensors and electronics. We first characterized the material's strain sensing characteristic by measuring the resistance change of a 37.6 mm × 4.6 mm × 2 mm piece with applied tensile strain (Fig. 3e) and pressure (Supplementary Fig. 14). The gauge factor (the slope of the linearly fitted curve of the relative resistance change vs. strain) of the material decreased with the SWCNT concentration, which resulted from the denser entanglement of SWCNTs at higher concentration that induced higher conductivity and lower strain-dependent sensitivity. It should be noted that our SWCNT/BBE possesses a good pressure sensitivity even in a small range (0–2 kPa) because of its ultralow Young's modulus[54]. Then, the material durability was characterized through 2500 cycles of loading-unloading tensile testing. The experimental results show relatively small decays of the relatively resistance and stress sensitivity (Fig. 3f and Supplementary Fig. 15), indicating good durability for long-term use.

Another factor that could affect the material's long-term use is its environment stability. We characterized the conductive SWCNT/BBE by storing it in an acid solution (HCl, pH = 2), an oxidative solution ($H_2O_2$, 30 wt%), and the ambient environment, all for 30 days. Because our BBE contains no water and organic solvents that are prone to evaporation and the SWCNTs have higher stability over metallic nanofillers (e.g., copper or silver nanowires) under oxidative and corrosive conditions[39], the material showed no obvious resistance change during the 30-day storage (Fig. 3g, Supplementary Fig. 16). In comparison, commercial conductive hydrogels (3M, Red Dot) showed a significant resistance increase after sitting in ambient environment for only 4 h (Fig. 3g). In addition to the confirmed sensitivity, durability, and stability of our SWCNT/BBE, its usage as wearable sensors and electrodes also requires a certain level of adhesion property of the

material on different substrates. We conducted lap shear testing of the SWCNT/BBE on the same five types of material surface and measured even higher shear strength values than those of pure BBE (Fig. 3h and Supplementary Fig. 17). The higher shear strength of the SWCNT/BBE over the pure BBE resulted from the higher Young's modulus (inducing less material deformation at the adhesion interface; see Supplementary Fig. 17) and the higher surface roughness (causing a larger total contact area) of the SWCNT/BBE[55]. With its adhesion property, the SWCNT/BBE can conformally adhere to skin and lift small objects (Supplementary Fig. 18 and Supplementary Movie 2). Lastly, we studied the cytotoxicity of the pure PDMS BBE and conductive SWCNT/BBE (0.4 wt%) through culturing human dermal fibroblasts on the material surface. After 24-h culture, the fibroblasts demonstrate 99%, 77%, 79%, and 64% viability (Supplementary Fig. 19) for pure BBE, 0.2 wt%, 0.4 wt%, and 0.6 wt% SWCNT/BBE, respectively, indicating relatively low cytotoxicity of the materials. The lower viability of the SWCNT/BBE could be attributed to the addition of the SWCNT to the BBE. We hypothesize that the commercially purchased SWCNT may contain proprietary compounds that could affect the observed results for cytotoxicity. A future investigation will be conducted to determine the presence of potential compounds existing in the SWCNT/BBE.

## Patterning of BBEs

For fabrication of ultrasoft electronics from conductive SWCNT/BBEs, a key technical challenge is to integrate materials (i.e., insulators and conductors) into monolithic devices through a simple yet efficient patterning strategy. To preserve the ultra-softness of the BBE to the maximum extent in the fabrication process as well as to increase the patterning capabilities, we employed two patterning methods here: laser cutting and 3D printing. Electronics with single-layer (2D) conductive SWCNT/BBE patterns can be fabricated by laser-cutting SWCNT/BBE into conductor patterns and then encapsulating them with pure BBE as insulators (Supplementary Fig. 20). Covalent bonds are formed between the SWCNT/BBE and pure BBE during the crosslinking process of the pure BBE insulator thanks to their shared monomers, crosslinkers and thermal initiators. The monomers and crosslinkers could diffuse into the cured SWCNT/BBE and form an interpenetrating polymer network at the interface between two BBEs upon crosslinking[56]. As a result, the samples prepared by this curing method show a higher shear strength and better stretchability compared to those of samples prepared by physically attaching (Supplementary Fig. 21 and Supplementary Movie 3). The patterning resolution of the laser cutting method was determined to be 0.6 mm. The Young's modulus of the laser-cut SWCNT/BBE (0.4 wt%) was measured to be 5.05 kPa ($N$ = 3), showing no significant change comparing to that of the pristine SWCNT/BBE. These results show that the laser cutting method not only provides good patterning resolution for single-layer SWCNT/BBE but also preserves the material's ultra-softness.

For electronic devices with complex conductor pathways and/or multilayer designs, 3D co-printing of conductive SWCNT/BBE and pure BBE becomes a promising option. There are only a few studies on 3D printing of non-conductive pure BBEs, and the preparation of BBE inks usually involves sophisticated synthesis[43,44,57]. Also, no 3D printing of conductive BBEs has been demonstrated for construction of integrated electronics. To adjust the printability of the conductive SWCNT/BBE, we prepared SWCNT/BBE inks by mixing 0.2 wt%, 0.4 wt%, or 0.6 wt% SWCNT with the BBE precursor. Similarly, the non-conductive BBE ink was prepared by vigorously mixing 4.5 wt% fumed silica with the BBE precursor. We found that the addition of SWCNTs effectively increased the viscosity of the BBE precursor, which is in good agreement with a previous study[58] (Supplementary Fig. 22a). With SWCNTs as the thixotropic agent, the SWCNT/BBE inks exhibit a typical shear-thinning behavior beyond a crossover point (~46% strain), which is defined by the storage modulus ($G'$) and loss modulus ($G''$) of the material (Supplementary Fig. 22b). The increases in the

viscosity and modulus of the SWCNT/BBE ink with the mixing concentration (0.2 wt%, 0.4 wt%, and 0.6 wt%) indicate the enhanced interactions among SWCNTs in the material that result in the superior shear-thinning behavior. The shear-recovery experiments further confirmed this shear-thinning behavior. The SWCNT/BBE inks at 0.4 wt% and 0.6 wt% concentration both revealed a fast and repetitive transition from the solid-like state at 1% strain to the viscoelastic state at 50% strain, showing their ability to quickly solidify after extrusion. The SWCNT/BBE ink at 0.2 wt% concentration shows a low viscosity and modulus in rheology measurements due to insufficient interactions among SWCNTs (Supplementary Fig. 22), thus limiting its recovery and solidification abilities for 3D printing (Supplementary Fig. 23a). Therefore, the SWCNT/BBE inks at 0.4 wt% and 0.6 wt% concentrations can be patterned by 3D printing (Supplementary Fig. 23b, c).

The non-conductive pure BBE has a low viscosity not compatible with 3D printing. Fumed silica is a commonly used thixotropic agent for hydrophobic polymers like PDMS, which can form a silica-silicone-silica 3D network in the polymer and thus enable the shear-thinning behavior of the material[59]. We used fumed silica as the additive to prepare the non-conductive BBE ink. Similar to adding SWCNTs to BBE, introducing fumed silica into BBE will increase the Young's modulus and hysteresis of the composite (Supplementary Fig. 24), and a trade-off is needed between the printability and ultra-softness of the silica/BBE. To this end, we tested silica/BBE inks at 4.5 wt% and 6.5 wt% concentrations and finally chose the one at 4.5 wt% for 3D-printing as it provides proper shear-thinning and shear-recovery behaviors (Supplementary Fig. 25), and a sufficiently low modulus (3.89 kPa).

## Laser-cut strain sensors

The patterning methods we developed allow the integration of BBE insulators and conductors into ultrasoft and ultrasensitive sensors that can be used in applications such as soft robotics and wearable sensing. Recently, significant efforts have been made to develop soft robots or wearable devices with strain sensors for position and force feedback. These strain sensors, typically made from polymers or liquid metals, have been co-fabricated or post-integrated into soft robots and wearables[60,61]. However, the softness of the polymer-based sensors is usually higher than that of the structural material (e.g., Ecoflex and PDMS) of soft robots, thus imposing additional mechanical constraints on the robotic actuation. This issue also exists in wearable sensing, where softer materials are preferred to provide better conformity and less constraints on human body[62]. Liquid metal, as the strain sensing material, is ultrasoft, but liquid metal-based strain sensors usually require elastomer-based microchannels to pattern liquid metal conductors and thus impose additional mechanical constraints to a soft robot or human body[63]. In addition, the typical elastomers (e.g., Ecoflex and PDMS) used in stretchable strain sensors do not have inherent adhesion to different material surfaces, and require additional adhesives for integration onto a robot or human body.

Thanks to the ultra-softness and adhesion property of the SWCNT/BBE, we hypothesize that our SWCNT/BBE-based sensors can be easily integrated into a soft robot and impose no obvious impact on the robotic actuation. To validate it, we fabricated an ultrasoft strain sensor with a single-layer SWCNT/BBE pattern through the laser cutting method (Fig. 4a, b and Supplementary Figs. 26a, b) and then attached it to an Ecoflex-based pneumatic actuator based on the sensor's own adhesion to Ecoflex (Fig. 4c). The 0.4 wt% SWCNT/BBE was used for sensor fabrication to balance the material's conductivity and softness. We also laser-cut a SWCNT/Ecoflex-based strain sensor with the same dimensions as the SWCNT/BBE-based one (Supplementary Fig. 26c), and compared the mechanical impact of the two sensors on the pneumatic actuator. As shown in Fig. 4c-d and Supplementary Movie 4, bending angles of the actuator with and without the SWCNT/BBE sensor show no obvious difference at different actuation pressures, while the attachment of the SWCNT/Ecoflex sensor significantly

reduced the bending angle of the actuator because of the large mechanical constraint imposed by the sensor. The relative resistance change of the SWCNT/BBE sensor with the actuation pressure shows a linear response (Supplementary Fig. 26f).

We further apply the laser-cut strain sensor to wearable sensing on human body, where the sensor was conformally adhered to the knuckle and fingertip through its adhesion (Fig. 4b and Supplementary Figs. 26g, h). By simply contacting the finger with the strain sensor, the sensor can attach to the skin and showed no device detachment when bending the finger (Supplementary Movie 5). In comparison, the SWCNT/Ecoflex-based strain sensor easily detached from the skin due to the poor adhesion of Ecoflex (Supplementary Figs. 26d, e). When bending the finger or touching a petri dish, the SWCNT/BBE strain sensors adhered on the knuckle and fingertip showed a rapid resistance change (Supplementary Fig. 26h), confirming their good on-body sensing performance. The low Young's modulus of our SWCNT/BBE even allows us to make wearable electronics for soft-bodied animals such as the hornworm. The hornworm body has a much lower modulus (37.7 kPa[64]) than that of epidermis skin (~1 MPa[65]). We hypothesize that the SWCNT/BBE could be used as a sensor to monitor the physical movements of soft-bodied animals due to its matched softness and satisfactory adhesion. For this application, one requirement is that the SWCNT/BBE should have no obvious physical constraints to the hornworm. Thus, we attached our SWCNT/BBE on the back of a hornworm (Fig. 4e), and recorded its crawling movements. As a comparison, the movement of the same hornworm without attaching the SWCNT/BBE was also recorded. We used the crawling speed (defined as: distance of a single crawling/time) and normalized change of the body length (defined as: body length after each crawling $L$/initial body length before a crawling $L_O$) to study the effect of attaching the SWCNT/BBE on the hornworm. The results show that both the crawling speed and body length change are not affected by attaching the sensor, indicating that our SWCNT/BBE has no obvious mechanical constraints to the physical movements of the hornworm (Fig. 4f, g, Supplementary Fig. 27a, and Supplementary Movie 6). In comparison, a piece of the PDMS Sylgard 184 (with the same dimensions as the SWCNT/BBE sensor) will easily detach from the hornworm body due to high stiffness and limited adhesion (Supplementary Fig. 27b and Supplementary Movie 6). Then, we recorded the resistance change of the SWCNT/BBE sensor with the crawling of the hornworm, and the long-term electrical response shows that our SWCNT/BBE sensor can easily record the movements of the hornworm without detaching (Fig. 4h and Supplementary Movie 7). The capability of conducting demonstrations with soft-bodied animals could possibly inspire research on biomechanics[66] and robot designs[67]. These results highlight the great potential of our conductive SWCNT/BBE as an ultra-compliant and sensitive material for soft robotic and wearable sensing.

## 3D-printed touch pad

The 3D-printing method allows us to fabricate multilayer structures of BBE-based electronics that the laser cutting method can hardly do. We 3D-printed a multilayer BBE-based touch pad consisting of 5 × 5 touch sensors. As shown in Fig. 4e, the touch pad includes top and bottom layers of orthogonally arranged sensing strips of conductive SWCNT/BBE, which are separated by a middle spacer layer of non-conductive pure BBE. The two sensor layers were 3D-printed with 0.4 wt% SWCNT/BBE, and the middle spacer layer and the top and bottom insulator layers were 3D-printed with 4.5 wt% fumed silica/BBE. The top insulator layer can also be spray coated with pure BBE precursors in cases where sensitivity is the first priority (Supplementary Fig. 28). The signal readout circuit is schematically shown in Supplementary Fig. 29. The crisscross areas of the top and bottom sensing strips form 25 touch sensors with a 1.5 mm pitch. When pressing each sensor, the resistance values of the two corresponding sensing strips on the top and bottom layers will change simultaneously. These simultaneous resistance

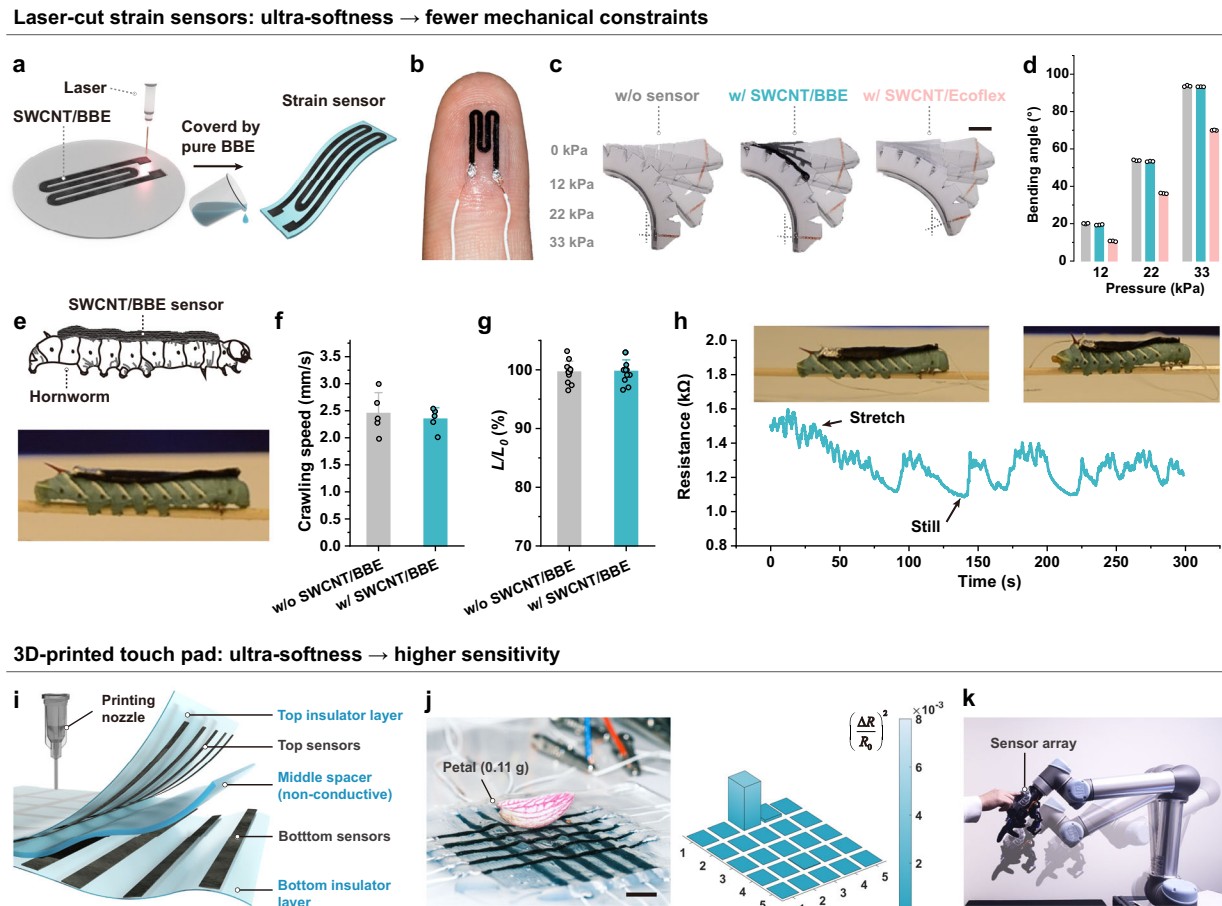

**Fig. 4 | Laser-cut strain sensors and 3D-printed touch pad based on ultrasoft BBEs. a** The schematic of the fabrication process of laser-cut strain sensors. **b** Photograph of the laser-cut strain sensor adhered on the fingertip of a human hand. The strain sensor can easily attach on human body due to its self-adhesive property. **c** Overlaid photographs of a pneumatic actuator (inflated at different pressure levels) without sensor, with a SWCNT/BBE strain sensor, and with a SWCNT/Ecoflex strain sensor. The inflation pressure levels were kept the same for the three groups. The scale bar is 1 cm. **d** Bending angles of the pneumatic actuators under 12 kPa, 22 kPa, and 33 kPa of inflated pressure in **c**: without sensor (gray), with a SWCNT/BBE strain sensor (blue), and with a SWCNT/Ecoflex strain sensor (pink).

**e** The schematic and photograph of a hornworm attached with the SWCNT/BBE sensor. **f** The crawling speed and **g** normalized length change of the hornworm with and without the SWCNT/BBE sensor. **h** The resistance change of the SWCNT/BBE sensor with the crawling of the hornworm. The insets show photographs of the hornworm at the two different states (left side photo: the hornworm body was stretched; right side photo: the hornworm kept still). **i** Expanded schematic of the printed touch pad. **j** The sensory response for placing a petal of the orchid on the touch pad. The scale bar is 5 mm. **k** Human–machine interaction through touching the robotic e-skin on the robot hand. Error bars denote the standard deviation of the measurements.

changes of the top and bottom sensing strips indicate the location and magnitude of the touch force. This multilayer crisscross sensor array design enables the integration of strain sensing elements at a high density, thus allowing touch sensing with a high spatial resolution. We calibrated the touch pad with 3D-printed top and bottom insulator layers and measured the relative resistance change of each force sensing strip as a function of touch force. The calibration curves (Supplementary Figs. 30a, b) indicate a measurement range of 0–0.1 N for each touch sensor. Based on three sigma limits statistical calculation (Supplementary Fig. 30c), the limit of detection of the touch pad was determined to be 0.00029 N. These data show that the touch pad is ultrasensitive to gentle touches because of the ultra-softness of the BBE materials.

To visualize the response of each touch sensor, we display the product of the relative resistance change of the top and bottom sensing strips of each sensor on the computer screen through a Matlab graphic user interface. We demonstrate that a touch pad (with spray-coated top insulator layer) can sense the location of small and light objects (Fig. 4f, Supplementary Fig. 31, and Supplementary Movie 8), such as a petal (0.11 g), orchid (0.96 g), blackberry (4.8 g), blueberries (1.72 g and 0.58 g), and butterfly specimens (0.55 g and 0.66 g). We also

attached a touch pad (3D-printed top insulator layer) to a robotic hand (mounted on a robotic arm) as a sensory electronic skin (e-skin), and employed it for human-machine interaction. To facilitate human interactions with the touch pad, we defined the touch pad into functional areas (each with a similar size to that of a fingertip) to receive human finger touches and generate control commands for the robotic arm accordingly (Supplementary Fig. 32a). In this way, we were able to control motions of the robot arm through the sensation of the e-skin (Fig. 4g, Supplementary Fig. 32b, and Supplementary Movie 9). Because of the high sensitivity of the BBE-based e-skin, the robot arm can sense and respond to gentle contact with a light object such as a ping-pong ball (Supplementary Fig. 32c) or a non-contact air blow from a compressed-air can (Supplementary Fig. 32d). These demonstrations showcase another application venue of our SWCNT/BBE for constructing ultrasoft e-skins with high sensitivity.

### 3D-printed electrodes for ECG recording
Besides sensor applications, the satisfactory conductivity and ultra-softness of the SWCNT/BBE also make it suitable for fabricating wearable electrodes for on-body electrophysiological measurement (Fig. 5a). We prepared circular electrodes of SWCNT/BBE at 0.2 wt%,

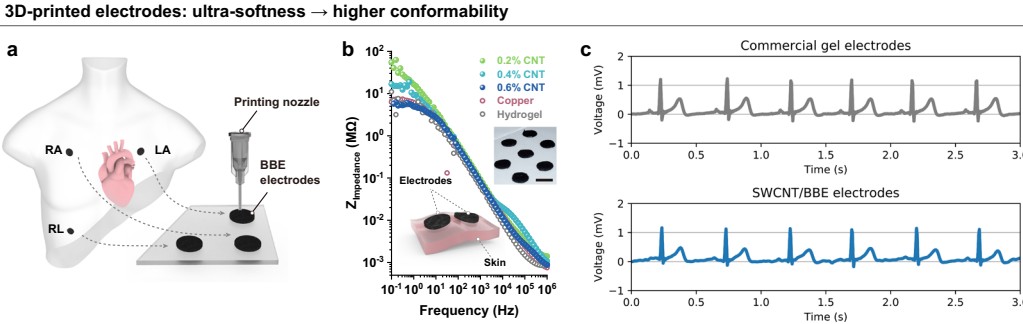

**Fig. 5 | 3D-printed SWCNT/BBE electrodes for ECG recording. a** Schematic of the ECG measurement. RA, LA, and RL represent the locations of right arm electrodes, left arm electrodes, and right leg electrodes, respectively. The three ECG electrodes were 3D printed with 0.6 wt% SWCNT/BBE. Dimensions are not to scale. **b** Bode plot of skin impedance measurements using SWCNT/BBE, copper sheet, and hydrogels as electrodes on the skin. The insets show a schematic of the measuring principle, and a photograph of 3D printed SWCNT/BBE electrodes. The scale bar is 2 cm. **c** Experimental data of ECG signals measured by commercial gel electrodes and SWCNT/BBE electrodes.

0.4 wt%, and 0.6 wt% concentrations, and attached two electrodes of the same kind onto human skin for electrical impedance measurement. For comparison, we fabricated circular electrodes made from copper film and hydrogel for skin impedance measurement on the same locations of the human skin (see Methods for experimental details). As shown in Fig. 5b and Supplementary Fig. 33, the impedance data from the three types of SWCNT/BBE electrodes reveal an overall decreasing trend with the SWCNT concentration, because a higher SWCNT concentration leads to a lower bulk resistance of the electrode. The impedance data from the 0.6 wt% SWCNT/BBE electrodes are even slightly lower than those from the copper electrodes in the low frequency range (<3 Hz), which could be attributed to the lower contact resistance at the interface of the SWCNT/BBE electrode and the skin because of the high electrode conformity. The impedance level of using the 0.6 wt% SWCNT/BBE electrodes is similar to that of using the hydrogel electrodes, indicating that our SWCNT/BBE electrode has comparable electrical performance to the gold-standard hydrogel electrode. This result confirms the feasibility of applying our solvent-free SWCNT/BBE to wearable electrodes, which could avoid the dehydration problem of the conventional hydrogel electrode. Moreover, we can easily fabricate the 0.6 wt% SWCNT/BBE electrodes with 3D printing method (Fig. 5b) and perform ECG measurements on a human body. By placing three SWCNT/BBE electrodes at the right arm position (RA), left arm position (LA), and right leg position (RL) (Fig. 5a), we were able to measure the ECG signal and compare it with that from commercial hydrogel electrodes (on the same human subject). From Fig. 5c, one can see that ECG signals can be successfully measured by our SWCNT/BBE electrodes, and show a similar pattern with the one measured by the commercial hydrogel electrodes. In addition, we overlayed the two patterns (Supplementary Fig. 34) and one can observe the P waves, QRS complex, and R waves from both patterns[68]. The amplitude and wave durations (e.g., PR interval and QRS duration) recorded from the SWCNT/BBE electrodes have few differences compared to those measured by the commercial gel electrodes. These results further indicate that the SWCNT/BBE could be a candidate for physiological measurements.

## Discussion
We have developed a conductive SWCNT/PDMS BBE with ultralow Young's modulus (2.98–10.65 kPa), satisfactory conductivity (2.68–13.78 S/m), solvent-free property, excellent environmental stability, inherent adhesion, and good biocompatibility. The nature of the ultra-softness and good conductivity of SWCNT/BBE derives from the unentangled bottlebrush polymer networks and SWCNTs that percolate in the polymer networks, thus leading to the development of ultrasoft electronics with soft tissue-matched softness. The conductive

SWCNT/BBE was successfully patterned by laser cutting and 3D printing methods and further integrated into BBE-based ultrasoft electronic devices such as strain sensors and wearable electrodes. We demonstrated the application of the conductive SWCNT/BBE for soft robotics, wearable strain sensing, human-machine interaction, and wearable electrophysiological measurement. Although more desirable properties, i.e., higher conductivity, and the chemical and processing space of such materials and devices, can be improved and developed, our conductive SWCNT/BBEs represent a promising material system for developing ultrasoft electronics that can find important applications in many areas.

## Methods
### Materials
The monomer monomethacryloxypropyl terminated poly-dimethylsiloxane (MCR-M11, average molecular weight 1000 g/mol) and the crosslinker methacryloxypropyl terminated poly-dimethylsiloxane (DMS-R22, average molecular weight 10,000 g/mol) were purchased from Gelest and purified using basic alumina columns to remove inhibitor. Specifically, a thin layer of cotton was first loaded at the bottom of a 10 ml syringe to avoid alumina leaking to the purified monomers or crosslinkers. A tube was placed at the outlet of the syringe to collect the purified materials. Next, 5 ml basic alumina was loaded into the syringe as the column. The monomers or crosslinkers were then added into the syringe to slowly pass the column until all the purified liquids were collected into the tube. The purified monomers and crosslinkers were stored at 2–4 °C before use. The thermal-initiator 2,2′-Azobis(2-methylpropionitrile) (AIBN, 440190-25G, 98%) was purchased from Sigma-Aldrich and used as received. The high-purity SWCNT (≥80%) were purchased from TUBALL. The fumed silica (AEROSIL R711) was obtained from EVONIK. Hydrogen peroxide solution ($H_2O_2$, H1009-100ML, 30 wt% in water), monomers acrylamide (AAm, 800830, ≥99%), crosslinkers N,N′-methylenebisacrylamide (MBAA, 146072-100G, 99%), and photo-initiators 2-hydroxy-4′-(2-hydroxyethoxy)−2-methylpropiophenone (Ir2959, 410896-50G, 98%) were purchased from Sigma-Aldrich and used as received. Hydrochloric acid (HCl333.500, 36.5%-38.0%) was purchased from BioShop. Acrylic acid (AAc, 99.5%) was purchased from Acros Organics.

### Preparation of pure PDMS BBE
Methacrylate-functionalized PDMS monomer (MCR-M11), crosslinker (DMS-R22), and azobisisobutyronitrile (AIBN, 2 mol% of PDMS monomers and crosslinkers) were added into a tube. The molar ratio of the PDMS monomer and crosslinker was set at 600:1, 900:1, or 1200:1. Then the solutions were mixed by a vortex mixer with a speed of 3000 rpm for 10 minutes, and the precursors were stored at 2–4 °C

before use. The elastomers were prepared by transferring the precursors into the Teflon mold and curing at 80 °C under $N_2$ overnight.

## Preparation of SWCNT/PDMS BBEs

The PDMS BBE precursors with the crosslinking ratio (molar ratio of MM:CL) of 1200:1 were prepared and well-mixed first. Next, 0.2 wt%, 0.4 wt%, or 0.6 wt% of SWCNT were added into the precursors and mixed for 30 min by a customized ball miller. The mixture was cooled at 4 °C after every 4 min of mixing. Then the mixture was transferred into the Teflon mold and kept under vacuum overnight to remove any bubbles in the mixture. Finally, the SWCNT/BBE was cured in the mold at 80 °C under $N_2$ overnight.

## Preparation of ionic hydrogels, PDMS Sylgard 184, and Ecoflex 00-10

The ionic PAAm-AAc hydrogels were synthesized by a one-pot method. Briefly, 1.8 g (0.025 mol) AAm, 0.9 g (0.0125 mol) AAc, 0.042 g Ir2959 (0.5 mol% of AAm and AAc), 241 μL (0.083 mol% of AAm and AAc) MBAA (20 mg mL$^{-1}$ in DI water), and 10 mL DI water were added in a tube. The mixture was mixed by a vortex mixer with a speed of 3000 rpm for 5 min and then transferred into the Teflon mold and photopolymerized under UV (365 nm, 10 mW cm$^{-2}$) for 1 h. The cured hydrogels were sealed in Ziploc bags and stored at 4 °C before measuring. PDMS Sylgard 184 was prepared by mixing part A and part B with the weight ratio of 10:1 and cured under 80 °C for 3 h. Ecoflex 00-10 was prepared by mixing part A and part B with a weight ratio of 1:1 and cured under 80 °C for 1 h.

## Mechanical testing

The cyclic uniaxial tensile stress–strain response of the elastomers was measured with a customized tensile test machine (with a 0.2 N load cell, S100 Strain Measurement Devices). The ISO 37 - Type 4 standard was adopted to conduct the tensile test. The BBEs were molded or cut into a dumb-bell test piece, with 2 mm width of narrow portion, 12 mm length of narrow portion, and 6 mm width of ends. The traverse speed of the moving grip was 50 mm min$^{-1}$ (strain rate: 0.070 s$^{-1}$). The speeds of 25 mm min$^{-1}$ (strain rate: 0.035 s$^{-1}$) and 100 mm min$^{-1}$ (strain rate: 0.14 s$^{-1}$) were also used to characterize the elasticity of BBEs. Each sample was subjected to 20 repeated stretching-releasing cycles or 1000 repeated cycles at the maximum strain of 50%, 100%, 200%, or 400% to characterize the long-term durability. The Young's modulus of the pure PDMS BBE was determined by the fitting model. For SWCNT/BBEs, the Young's modulus was determined by linear fitting the stress-strain curve at the strain range of 0–10%.

The adhesion shear strength of the pure BBE with the crosslinking ratio of MM:CL = 1200:1 was tested under dry conditions by the standard lap-shear test with the same tensile test machine (with a 0.2 N load cell). All the elastomer samples (PDMS BBEs, PDMS Sylgard 184, and Ecoflex 00-10) and porcine skin were adhered with a stiff thin-film tape by cyanoacrylate glue (Krazy Glue) as the rigid backing. The adhesion area of the materials has a width of 13.8 mm and a length of 10 mm. In all cases, the tensile speed was 50 mm min$^{-1}$. The shear strength was determined by dividing the maximum force by the adhesion area.

## Resistance and conductivity testing

The resistance of SWCNT/BBE with the dimension of 37.6 mm × 13.8 mm × 2 mm (length × width × thickness) was measured by a source meter (Keithley 2614B, Keithley Instrument Inc.). The composite of Ag flakes and liquid metal EGaln was applied at the interface of the electrodes and elastomer surface to reduce the contact resistance. The conductivity $\sigma$ was calculated by the equation $\sigma = l/RA$ using the length ($l = 37.6$ mm) and cross-sectional area ($A = 27.6$ $m^2$). For the measurement of sensitivity, the copper film tapes were adhered to the grip of the tensile test machine as electrodes. A mixture of Ag flakes and liquid

metal EGaln (10 wt% Ag flakes and 90 wt% EGaln) was used at the probe-elastomer interface to effectively reduce the contact resistance (Supplementary Fig. 35). The resistance change of the SWCNT/BBE upon stretching or compressing was measured by the source meter. The applied force and displacement of the elastomer were measured by the tensile test machine.

## Electrochemical impedance spectroscopy analysis

The AC-impedance spectra of the SWCNT/BBE with the dimension of 37.6 mm × 13.8 mm × 2 mm (length × width × thickness) was measured by an electrochemical workstation (Autolab PGSTAT302N, Metrohm), with the testing frequency ranging from $10^5$ Hz to 0.1 Hz. The measuring sinusoid amplitude was 10 mV with no bias applied. For on-skin impedance measurement using SWCNT/BBEs with different SWCNT concentrations as soft electrodes, the elastomers prepared by molding and then cut by a puncher into circular samples with the dimension of 8 mm in diameter and 2 mm in thickness. Hydrogels (2 mm in thickness) and copper film (0.07 mm in thickness) were cut into the same circular shape to compare with the SWCNT/BBE electrodes. The electrodes were placed on the skin with the spacing of 1 mm, and the AC-impedance was measured by the same setting above. In all cases, the composite of Ag flakes and liquid metal EGaln was used at the interface of material surfaces and probes of the electrochemical station to reduce the contact resistance.

## Environment stability test

Three groups of conductive SWCNT/BBEs with 0.4 wt% SWCNT were stored in ambient air, hydrochloric acid solution (pH = 2), and hydrogen peroxide solution (30 wt% in water), respectively. Each group has three samples with the dimension of 37.6 mm × 13.8 mm × 2 mm (length × width × thickness). The samples stored in air and the hydrochloric acid solution were kept at room temperature (23 °C), and the samples stored in the hydrogen peroxide solution were kept under 4 °C. After storage for 10 days, 20 days, and 30 days, the resistance of each sample was measured in the same way as the resistance and conductivity test. Before measuring the resistance, the samples stored in solutions were dried under 60 °C for an hour to remove any solutions on samples and cooled to room temperature for resistance measurements.

## HDF culture and viability assay

Human Dermal Fibroblasts (HDF) were cultured in Dulbecco's Modified Eagle's Medium (DMEM, Gibco Cat. 11995) supplemented with 10% fetal bovine serum (Gibco). Pure bottlebrush elastomers and bottlebrush elastomers with 0.2 wt%, 0.4 wt%, or 0.6% CNT were prepared in the wells of 96 well plates (Greiner Bio-One). BBE and SWCNT/BBE containing wells were coated with 0.1% gelatin (STEMCELL Technologies). 15,000 HDF cells were seeded into each BBE or SWCNT/BBE containing well, as well as blank control wells. After 24 h, viability was determined by incubating with 1:5 volume of MTS reagent (Promega, G3582) for 90 minutes, then reading absorbance at 490 nm using VersaMax 190 plate reader (Molecular Devices). Absorbance values were normalized to blank control wells.

## Rheological characterization

Rheological characterization of pure PDMS BBEs and 3D printing BBE inks was performed on a rheometer (DHR-3, TA Instrument) with a 40 mm diameter parallel plate steel geometry. All measurements were conducted at 25 °C with a 1 mm gap. The storage modulus of pure BBEs was measured by sweeping the angular frequency from 0.01 rad/s to 100 rad/s at the strain of 1%. For inks, the viscosity was measured by the flow ramp mode with a ramped shear rate from 0 to 50 1/s. The strain sweep (0.1–100%) was performed by the oscillation amplitude mode at 1 rad/s. The shear recovery experiments were conducted by the

oscillation time mode at 1 rad/s, alternating 200 s at 1% strain, with 200 s at 50% strain.

## 3D printing of SWCNT/BBE-based devices

The 3D-printing inks of conductive SWCNT/BBE were prepared by the same method for preparing the SWCNT/BBE precursors, where 0.4 wt % (for sensor applications) and 0.6 wt% (for electrodes applications) of SWCNT were used as conductive fillers and also thixotropic agents. The non-conductive pure BBE inks were prepared by mixing 4.5 wt% of fumed silica with pure BBE precursors. After well-mixed, the non-conductive inks and conductive inks were loaded into two 5 cc syringes, respectively, and then centrifuged at the speed of 264 g for 10 s to remove any bubbles in the inks. The prepared inks were kept at 4 °C before 3D printing.

3D printing of the non-conductive pure BBE and the conductive SWCNT/BBE was conducted by a direct-ink-writing 3D printer (Engine SR, Hyrel 3D) with the reservoir-based printing heads (SDS 5, Hyrel 3D) loaded with a 5 cc syringe and a 0.58 mm nozzle. The motion of the printer was controlled via GCode, which was converted by a commercial software (Slic3r) based on CAD-generated printing patterns. The 3D-printed device was polymerized at 80 °C under vacuum overnight for complete curing. For the touch pad device, the composite of Ag flakes and liquid metal EGaIn was used to connect the ends of SWCNT/BBE strips with copper wires to reduce the contact resistance, and Ecoflex was further applied to seal the connection joints. In testing and demonstrations of the touch pad, the top surface of the touch pad was spray coated with the solution of the universal mold release (Smooth-On, Inc.) to reduce adhesion.

## Human–machine interaction

The 3D-printed touch pad was attached to a robot hand (Youbionic) which was connected with a robot arm (UR5, Universal Robots, operated by ROS software) for human–machine interaction. The ten sensor strips of the touch pad were serially connected with ten resistors (10 kΩ) powered by a 5 V source voltage (Fig. S27). An Arduino microcontroller (Arduino Mega) was used to measure the voltage across the sensor strips. The output voltages were then transferred to Python to calculate and record the location of pressed points on the touch pad, thus generating a real-time command to trigger the motion of the robot arm.

## ECG recordings

The electrodes were 3D-printed with the SWCNT/BBE ink (0.6 wt%). After curing, the electrodes were fixed into the plastic cover of a commercial electrode (3 M, Red Dot), where a thin layer of liquid metal was deposited to reduce the contact resistance between SWCNT/BBE and Ag/AgCl electrodes in the plastic cover. Then, three SWCNT/BBE electrodes were attached to the right arm position, left arm position, and right leg position of a human chest, with the covered Ag/AgCl electrodes connecting three wires to a commercial ECG monitor chip (AD8232). An Arduino microcontroller (Arduino Mega) was used to power the chip and record the signal output from the chip. The Arduino was serially connected with Python, which was used to filter and save the real-time ECG signal. The participants were authors of this paper, and consent was obtained from research participants before conducting the experiments.

## Data availability

All of the data in this work are presented in the main text and the supplementary information. Additional data are available from the corresponding author upon reasonable request.

## Code availability

The custom code used in the demonstrations is available from the corresponding author upon reasonable request.

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

## Acknowledgements

The authors thank Siwan Park and Professor Edmond W. K. Young for providing human dermal fibroblasts, and Crystal Liu on her assistance on building the tensile test machine. The authors acknowledge financial support provided by the Natural Sciences and Engineering Research Council of Canada RGPIN-2022-05039 (X.L.), Natural Sciences and Engineering Research Council of Canada RGPIN-2021-03554 (H.T.),

Natural Sciences and Engineering Research Council of Canada RGPIN-2017-06374 (X.L.), Natural Sciences and Engineering Research Council of Canada RGPAS-2017-507980 (X.L.), Canada Foundation for Innovation JELF-38428 (X.L.), Canada Foundation for Innovation JELF-41743 (H.T.), and XSeed Grant from the University of Toronto (X.L., H.T.). A.L. thanks NSERC for a graduate scholarship (CGS-D).

## Author contributions

P.X., S.W., A.L., H.T., and X.L. designed the experiments, prepared the figures, and wrote the manuscript. P.X. and A.L. prepared the materials. P.X. and S.W. performed mechanical and electrical characterizations, 3D printing and demonstrations. H.-K.M. and X.H. designed and performed experiments of in vitro cytotoxicity. Z.Z. provided assistance on performing experiments with the robot arm. W.D. and Y.S. provided assistance on cell culturing.

## Competing interests

The authors declare no competing interests.
