## [Peer Review File · Nature Communications]

REVIEWER COMMENTS

Reviewer #1 (Remarks to the Author):

This work demonstrates ultrasoft and conductive elastomers using a composite system of PDMS-based bottle brush elastomers (BBE) and single walled carbon nanotubes (SWCNTs). Their BBE can be prepared just by mixing three materials, and thermal curing. The materials can show Young's Modulus lower than 1 kPa, and high hysteresis-free stretchability of 400%. When SWCNTs are incorporated, the materials showed modest conductivity of ~ 10 S/m while showing a low Young's Modulus of ~ 10 kPa. Furthermore, the authors demonstrate the uniqueness of their BBE/SWCNT composites by various applications including sensors for robotics and wearables. This manuscript is able to demonstrate a promising materials-based approach to realize ultrasoft electronics. Still, the following concerns should be addressed before I recommend the publication.

1. Young's Modulus is one of the important mechanical properties to modulate the softness of the materials. Still, geometries, including thickness, also significantly affect the softness of the material. Ultrathin polymer sheets or nanomeshes have shown very nice conformability to arbitrary surfaces. Although the authors claim "their limited stretchability and inherently higher modulus can result in unintended immunological responses and/or local damage to tissue" on Page 2, references 17 and 18 do not support this claim. A fair comparison of this approach with the authors' should be made.
2. Synthesis of PDMS-based BBE was recently reported in the following article. (Ref. 33, <https://www.nature.com/articles/s41467-022-28015-2>) The authors should compare their BBE with this.
3. Toxicity of SWCNT/BBE in Page 8 needs more detailed discussion. There is no evidence to show the chemical residuals in SWCNTs affect.
4. Although the authors criticize the previous work on BBE/SWCNT composite in Ref. 37, it is not clear why the authors' materials give better conductivity with smaller modulus. The authors should discuss this point. Perhaps only the materials preparation including ball milling gives this big difference.
5. Characterization of composites needs more clarification. For example, are SWCNTs, which usually form bundles, uniformly dispersed in the BBE matrix? Fig. S11 is not enough to show whether SWCNTs are uniformly mixed in BBE. SEM images in higher magnification are necessary. In addition, Cross-sectional SEM images would be helpful as most of the conducting paths shouldn't on the surface.
6. Recently, conducting polymer-based hydrogels attracts attention as ultrasoft and conductive materials. (e.g. <https://www.nature.com/articles/s41467-018-05222-4>) The authors should compare their work with them.

Reviewer #2 (Remarks to the Author):

The goal of this study was to prepare soft (tissue-mimetic) and conductive polymers. The group attempted this by synthesizing chemically crosslinked bottlebrush PDMS elastomers blended with single walled carbon nanotubes (SWCNT). The synthesis of the conductive materials (MCR-M11 PDMS bottlebrush with DMS-R22 crosslinker initiated by AIBN and blended with TUBALL SWCNT) is easy enough to follow and the material properties makes sense for the most part. Meaning, it is completely logical that the 1200:1 monomer:crosslinker ratio sample has the lowest modulus and the 600:1 monomer:crosslinker ratio has the highest modulus. I agree with authors' selection of 1200:1 monomer:crosslinker ratio sample for blending with the SWCNT due to the inherent softness of this material compared to the other options (900:1 & 600:1). The prepared materials were applied to two different fabrication methods: laser cutting and 3D printing. The authors demonstrated the capability of their materials in applications such as wearable sensing (pressure sensing/adhesion), soft robotics, and electrophysiological recording (ECG).

However, I question the novelty of the research, as it appears to me that the authors have simply taken one of many extensively reported recipes for bottlebrush elastomers and blended it with the TUBALL SWCNT which is exactly what has been recently reported in ref. 37. The authors repeatedly attempted to make correlations with little to no quantifiable data for backing (e.g., adhesion, ECG graph) and they directly contradict previously published mechanical and conductivity properties for almost identical materials. This paper offers nothing new in terms of novel chemistry, and the way the various applications are presented seems subjective and unconvincing. For these reasons, I would not recommend this paper for publication.

Other technical issues:

Fig. 2b: The strain rate selection is unclear, which questions the stiffness characterization procedure. The Young's modulus should be measured at the rubber elastic plateau, which should be verified by modulus measurements as a function of frequency. Note that samples with different crosslink densities demonstrate different frequency profiles. Also, the strain rate should be reported in s^{-1} (not in absolute units).

Line 425-426: With regard to the stiffness data in Figures 2 and 3, more details about the fitting model should be provided along with the used fitting parameters and their values.

On line 165, extremely high conductivity values (2.68, 7.08, and 13.78 S/m) are reported, which contradicts to the corresponding values reported in ref 37 using the same TUBALL SWCNT material. In ref 37, conductivity values of 0.01 S/m and 0.09 S/m were reported for SWCNT concentration of 0.25 and 0.51 wt % . Perhaps this discrepancy can be attributed using an alternative method for measuring conductivity.

Line 133-141: I do not agree with the claim to “strong adhesion” properties without at least meeting the shear strength of commonly used adhesive materials. Must be some benchmark to qualify for strong adhesion.

Line 222-224: The sentence “Covalent bonds are formed between the SWCNT/BBE 223 and pure BBE during the crosslinking process of the pure BBE insulator thanks to their shared monomers, 224 crosslinkers and thermal initiators,” seems out of place to me.

Lines 321-324: In addition to all the lightweight materials they use as a proof point for how sensitive the pressure sensors are, it would be good to determine what the minimum mass of detection is for the system.

Lines 347-349 & fig 5b: I am not convinced that the plot in figure 5b indicates that the impedance from the 0.6 wt% SWCNT/BBE electrodes are in good agreement with those of the hydrogel electrodes because all the samples have a similar profile.

Line 338: The 0.2, 0.4, and 0.6 wt % SWCNT/BBE mixtures are considered for biomedical applications. However, on lines 210-214, only the cell viability of 0.4 wt% is reported. If the 0.6 wt% is going to be used for biomedical applications, then it should be included in the biocompatibility testing.

Line 356-358: I do not understand the correlation of the two ECG's. The authors don't discuss any comparison. No proof point, they just show two ECG's and say they're correlated.

Reviewer #3 (Remarks to the Author):

dear authors:

I did enjoy reading your manuscript. It is useful and informative. The video files provided are useful and give light to the audience about what you did and difference between samples' behavior. What you claimed in the introduction based on your scenario is realized in your work. I guess some microscopic analyses would be useful, but even at the current form your works take credit for publication.

The methodology, results, and evidences provided as supplementary files are all professionally orchestrated. You also almost completely covered the area, but I think some review papers recently published on this field can be used in the introduction to emphasize classifications made on bottle brush polymers in terms of responsiveness, chemistry, etc. It would attract the general readers. What I propose is to better highlight the necessity of your work in the abstract and also in the introduction.

Regarding analyses, compare your works with similar advanced materials and systems available in the literature. Since your work is practically viable, make good use of patents as well. Technically, you can compare your results in that manner to emphasize the worth of what you did. Moreover, I'd expect a mechanistic explanation for the ending part, in a molecular level. Overall, I did not find something wrong to be mentioned, so you can publish your work after such minor corrections.

Reviewer #1 (Remarks to the Author):

This work demonstrates ultrasoft and conductive elastomers using a composite system of PDMS-based bottle brush elastomers (BBE) and single walled carbon nanotubes (SWCNTs). Their BBE can be prepared just by mixing three materials, and thermal curing. The materials can show Young's Modulus lower than 1 kPa, and high hysteresis-free stretchability of 400%. When SWCNTs are incorporated, the materials showed modest conductivity of ~10 S/m while showing a low Young's Modulus of ~10kPa. Furthermore, the authors demonstrate the uniqueness of their BBE/SWCNT composites by various applications including sensors for robotics and wearables. This manuscript is able to demonstrate a promising materials-based approach to realize ultrasoft electronics. Still, the following concerns should be addressed before I recommend the publication.

1. Young's Modulus is one of the important mechanical properties to modulate the softness of the materials. Still, geometries, including thickness, also significantly affect the softness of the material. Ultrathin polymer sheets or nanomeshes have shown very nice conformability to arbitrary surfaces. Although the authors claim "their limited stretchability and inherently higher modulus can result in unintended immunological responses and/or local damage to tissue" on Page 2, references 17 and 18 do not support this claim. A fair comparison of this approach with the authors' should be made.

Response:

Thank you very much for the kind suggestion. Yes, we agree with the reviewer's opinion. The materials' geometries and thickness will affect the applied stress on the biological

tissues. For instance, the ultrathin polymer sheets or nanomeshes are nice examples that can effectively reduce mechanical constraints and increase conformability to arbitrary surfaces ^{1,2}. The ultrathin sheets and nanomeshes provide a good solution to make soft materials by employing new fabrication technologies with existing polymers, while our bottlebrush elastomers (BBEs) achieve the softness by employing a new type of polymer. The realization of BBE-based ultrasoft electronics does not require the complex fabrication processes of ultrathin sheets and nanomeshes. In addition, the Young's modulus of BBEs is independent with the geometry or thickness of the materials, so the BBEs can be fabricated into different shapes and geometries while still maintaining the ultra-softness. This advantage could enable BBE-based devices with more application scenarios (e.g., wearable sensing and soft robotics).

To make it clear, we have added the following discussions in the revised main text on page 3:

“Ultrathin polymer sheets or nanomeshes are nice examples that can effectively reduce mechanical constraints and increase conformability to arbitrary surfaces ^{1,3,4}, but the ultrathin structures with in-plane patterns require complex device fabrication processes.”

2. Synthesis of PDMS-based BBE was recently reported in the following article. (Ref. 33, <https://www.nature.com/articles/s41467-022-28015-2>) The authors should compare their BBE with this.

Response:

Thank you very much for pointing this out. In general, there are three main approaches to synthesize bottlebrush polymers: grafting-from, grafting-to, and grafting-through methods ⁵. The article (Ref. 33) employed “grafting-to” approach to prepare the super-soft silicone elastomers, with mono-vinyl functional grafted onto the multi-hydride functional PDMS backbone to form bottlebrush side chains. In our work, we used “grafting-through” approach by directly polymerizing and crosslinking the reactive polymeric side chains into bottlebrush polymer networks. Both methods are simple and require only one reaction. The advantage of our method is that the grafting density is higher, and our Young's modulus is lower than that in Ref. 33 (0.63 kPa to 1.85 kPa versus 1.2 kPa to 7.4 kPa). The advantage of the method in the Ref. 33 is that their reaction is tolerant with oxygen and water. Thus, the selection of preparation methods of BBEs depends on the need of users. We want to achieve a lower Young's modulus for pure BBE so that we would be able to make the SWCNT/BBE composite softer. Thus, we selected the “grafting-through” method to obtain a softer BBE matrix.

To make it clear, we have added discussions of the preparation methods for pure BBEs on page 5:

“To this end, Vatankhah-Varnosfaderani ⁶ has proposed to use commercially available PDMS monomers to polymerize bottlebrush polymer networks by a “grafting-through”

approach. Hu et al. ⁷ recently reported a one-reaction method to simply prepare super-soft silicone elastomers. While compared to grafting side chains onto a polymeric backbone, the “grafting-through” method allows us to form bottlebrush polymers with higher grafting density and potentially provide a lower Young’s modulus.”

3. Toxicity of SWCNT/BBE in Page 8 needs more detailed discussion. There is no evidence to show the chemical residuals in SWCNTs affect.

Response:

Thanks for the suggestion. The reviewer is correct that there is no evidence for this statement. We initially included it as our hypothesis but we edited the manuscript text to properly reflect these observations. In the future, as these materials are used for biological applications, we will investigate this phenomenon.

We removed the sentence: “The reduced viability of the SWCNT/BBE could be caused by certain chemical residuals in the commercial SWCNT.”

We included the following discussions on page 9: “The lower viability of the SWCNT/BBE could be attributed to the addition of the SWCNT to the BBE. We hypothesize that the commercially purchased SWCNT may contain proprietary compounds that could affect the observed results for cytotoxicity. A future investigation will be conducted to determine the presence of potential compounds existing in the SWCNT/BBE.”

4. Although the authors criticize the previous work on BBE/SWCNT composite in Ref. 37, it is not clear why the authors' materials give better conductivity with smaller modulus. The authors should discuss this point. Perhaps only the materials preparation including ball milling gives this big difference.

Response:

Thanks very much for the comments and we are sorry for not describing the experiments in detail. In our measurement of the conductivity, we used Ag flakes and liquid metal EGeIn at the contact interface of the probes and materials (as shown in **Fig. R1-1a** below; **Supplementary Figure 35** in the manuscript). By using the liquid metal, one can effectively reduce the contact resistance ⁸. As a result, we can get the resistance of the material itself without involving the contact resistance. We conducted the conductivity measurements on the 0.4 wt% SWCNT/BBE sample, and the results show that the measured conductivity (5-7 S/m) by using the liquid metal would be much higher than that (0.01-0.04 S/m) of not using the liquid metal (**Fig. R1-1b**). In Ref. 37, we didn’t find any detailed information on the conductivity measurement. If not using the liquid metal to reduce the contact resistance, the measured resistance would be largely affected by the contact resistance induced by the measuring methods. We think that using the liquid metal is a more accurate way to determine the materials’ conductivity properties.

To make it clear, we have further clarified our measuring method on page 19:

“A mixture of Ag flakes and liquid metal EGaIn (10 wt% Ag flakes and 90 wt% EGaIn) was used at the probe-elastomer interface to effectively reduce the contact resistance (Supplementary Fig. 35).”

Compared with Ref. 37, our materials have a much lower modulus. In Ref. 37, the moduli of samples with 0 wt%, 0.25 wt%, and 0.51 wt% CNTs are 31 kPa, 66 kPa, and 140 kPa, respectively. While the moduli of our PDMS BBE samples with 0 wt%, 0.2 wt%, 0.4 wt%, and 0.6 wt% CNTs are 0.63 kPa, 2.98 kPa, 5.29 kPa, and 10.65 kPa, respectively. The modulus difference could be attributed to our different chemical components of the bottlebrushes and different crosslinking levels. As our priority was to keep the softness of the conductive BBEs in the soft tissue range (0-30 kPa), we selected a low crosslinking level (crosslinking ratio of MM:CL = 1200:1) to ensure the ultralow Young’s modulus (0.63 kPa) of pure PDMS BBE. Thanks to this ultrasoft material matrix, we were able to make the modulus range of SWCNT/BBEs fit within the range of soft biological tissues, and even the highest Young’s modulus (10.65 kPa) of our SWCNT/BBE is much lower than that of Ref. 37.

To make it clear, we have added discussions on page 7:

“This could be attributed to our different chemical components of the bottlebrushes and looser crosslinking networks.”

Fig. R1-1. **a** Photographs of the 0.4 wt% SWCNT/BBE without and with liquid metal as the interface between the probe and samples. **b** Measured conductivity of 0.4 wt% SWCNT/BBE with and without using the liquid metal.

5. Characterization of composites needs more clarification. For example, are SWCNTs, which usually form bundles, uniformly dispersed in the BBE matrix? Fig. S11 is not enough to show whether SWCNTs are uniformly mixed in BBE. SEM images in higher magnification are necessary. In addition, Cross-sectional SEM images would be helpful as most of the conducting paths shouldn't on the surface.

Response:

Thank you very much for the valuable comments. We have conducted additional SEM experiments based on the reviewer's suggestions. We quenched and broke our SWCNT/PDMS BBE samples in liquid nitrogen and took the cross-sectional SEM images (**Fig. R1-2**). One can see that in a higher magnification, there are CNT bundles formed in our samples. We thank the reviewer's comments for correcting this. We speculate that these bundles already exist in the commercial CNT raw materials as the pristine CNT cannot be well dispersed in the BBE precursor unless mixing and dispersing them in a ball milling machine. A more thorough or powerful mixing and dispersing method could possibly separate CNT from those bundles and increase the dispersing uniformity of CNTs. However, we found that the long-time ball milling or ultrasound treatment could generate a large amount of heat, which will affect the crosslinking of our elastomers. Thus, the time of the mixing process was controlled to prevent over-heating. To make it clear, we have supplied the SEM images in the revised supplementary files (**Supplementary Fig. 11**) and revised the description in the manuscript (Page 7):

“The scanning electron microscopy (SEM) images of the SWCNT/BBEs with different loading concentrations (0.2 wt%, 0.4 wt%, and 0.6 wt%) show the distributions of SWCNTs on the elastomer surfaces where the SWCNT density increased with the loading concentration. While the cross-sectional SEM images show that there are SWCNT bundles in the elastomer matrix (**Supplementary Fig. 11**), possibly derived from the pristine entangled SWCNTs.”

Fig. R1-2. SEM images at **a** 11000 \times and **b** 25000 \times of the cross-sectional surfaces of 0.4 wt% SWCNT/BBE samples.

6. Recently, conducting polymer-based hydrogels attracts attention as ultrasoft and conductive materials. (e.g. <https://www.nature.com/articles/s41467-018-05222-4>) The authors should compare their work with them.

Response:

Thank you very much for the suggestion. We agree with the reviewer that the conducting polymer-based hydrogels can also be promising tools to build ultrasoft electronics due to their low Young's modulus (2 – 32 kPa)⁹⁻¹². However, the high water content in hydrogels could limit their applications. The hydrogels could easily dehydrate when exposed to air and their conductivity might decay after long-term exposure. Thus, in page 2 we briefly discuss limitations of conductive hydrogels and also ionogels:

“Another approach is to use intrinsically stretchable materials (e.g., hydrogels^{9,10,13,14} and ionogels^{15,16}); these materials contain high water content or expensive ionic liquids that can diffuse, which limits the environments in which they can be employed.”

Besides, in **Fig. 3g** we used the environmental stability test to show the dehydration problem of hydrogels. It can be found that the hydrogels have a significant conductivity decay after exposed to air for only 4 h, while our BBEs can remain stable even after 30 days. In some application scenarios (e.g., wearable sensors in our demonstrations), the limited environmental stability could affect the electrical performance of the ultrasoft devices.

Reviewer #2 (Remarks to the Author):

The goal of this study was to prepare soft (tissue-mimetic) and conductive polymers. The group attempted this by synthesizing chemically crosslinked bottlebrush PDMS elastomers blended with single walled carbon nanotubes (SWCNT). The synthesis of the conductive materials (MCR-M11 PDMS bottlebrush with DMS-R22 crosslinker initiated by AIBN and blended with TUBALL SWCNT) is easy enough to follow and the material properties makes sense for the most part. Meaning, it is completely logical that the 1200:1 monomer:crosslinker ratio sample has the lowest modulus and the 600:1 monomer:crosslinker ratio has the highest modulus. I agree with authors' selection of 1200:1 monomer:crosslinker ratio sample for blending with the SWCNT due to the inherent softness of this material compared to the other options (900:1 & 600:1). The prepared materials were applied to two different fabrication methods: laser cutting and 3D printing. The authors demonstrated the capability of their materials in applications such as wearable sensing (pressure sensing/adhesion), soft robotics, and electrophysiological recording (ECG).

However, I question the novelty of the research, as it appears to me that the authors have simply taken one of many extensively reported recipes for bottlebrush elastomers and blended it with the TUBALL SWCNT which is exactly what has been recently reported in ref. 37. The authors repeatedly attempted to make correlations with little to no quantifiable data for backing (e.g., adhesion, ECG graph) and they directly contradict previously published mechanical and conductivity properties for almost identical materials. This paper offers nothing new in terms of novel chemistry, and the way the various applications are presented seems subjective and unconvincing. For these reasons, I would not recommend this paper for publication.

Response:

Thank you very much for the constructive comments. We respectfully clarify the novelty of our work below. In this work, we used commercially available PDMS monomers and crosslinkers to synthesize the ultrasoft bottlebrush elastomers (BBEs), and avoided complex chemical synthesis process. The development of ultrasoft BBEs has given us an opportunity to modify new functions and explore their applications, while the multi-step chemical synthesis involved in the previous work may increase the complexity of further modifications or fabrications. To provide a fair comparison, we discuss this in the revised manuscript on page 4 and also added a supplementary table to compare the different methods for preparing BBEs:

“Although the intricate tailoring of bottlebrush polymers enables new morphology^{17,18}, mechanical properties¹⁹, and crosslinking strategies^{20–22} for existing BBEs (Supplementary Table 1), complicated procedures, low overall yield and reproducibility

of the existing synthesis methods could hinder practical applications of ultrasoft elastomers.”

We have compared different methods of making BBEs in **Table R2-1** (supplied in the revised supplementary files: **Supplementary Table 1**) and concluded that the employed method in this work (using PDMS monomers and crosslinkers ⁶) is one of the simplest methods in literature. So one can easily modify the BBE (e.g., render the conductivity) and fabricate the BBE (e.g., adding fumed silica for 3D printing) without involving complicated chemical synthesis. Our simple synthesis method could make our materials more acceptable to non-experts, and potentially more applicable in different areas. With this simple method, we can avoid the multi-step synthesis of bottlebrush polymers in Ref. 37, and our SWCNT/BBE (2.98 kPa to 10.65 kPa) is also at least five times softer than those (66 kPa) in Ref. 37. Moreover, we systematically studied the elasticity, stretchability, environmental stability, cytotoxicity, and device integration methods of conductive BBEs, which are not involved in previous studies.

Table R2-1. Comparison of different methods of preparing bottlebrush elastomers.

Bottlebrush elastomers	Method	Side chains	Crosslinker	Commercially available	Young's modulus	Reference
PDMS BBE	Free radical polymerization	PDMS (MCR-M11)	PDMS (DMS-R18, DMS-R22)	All materials are commercially available	0.84 - 65.48 kPa	6,23–25
PDMS BBE	Hydrosilylation + platinum catalyst	PDMS (MCR-H21)	PDMS (DMS-H25)	All materials are commercially available	7.43 - 132 kPa	26,27
		PDMS (MCR-V21, MCR-V25)	PDMS-based crosslinker		1.2 - 7.4 kPa	7
linear-bottlebrush-linear elastomers	Atom transfer radical polymerization (ATRP)	PDMS, PMMA	Aggregation of linear polymer tails	No	2.1 - 155.7 kPa	6,28–30
PDMS-r-PEG elastomers	ATRP	PDMS, PEG	Isocyanate:hydroxyl (NCO:OH) and isocyanate:amine	No	1.8 - 27.8 kPa	22

			(NCO:NH ₂) coupling			
ROMP PDMS BBE	Ring-opening metathesis polymerization (ROMP)	PDMS	bis-benzophenone-based PDMS photocrosslinker	No	6.2 - 92 kPa	31
Poly(4-methylcaprolactone) (P4MCL) BBE	Ring-opening metathesis polymerization (ROMP)	P4MCL	4,4'-bioxepane-7,7'-dione	No	10 - 100 kPa	21,32
PDMS-stat-PEO BBE	Ring-opening metathesis polymerization (ROMP)	PDMS, PEO	bis-benzophenone-based PDMS photocrosslinker	No	32 kPa and 7.7 kPa	19

Also, we truly appreciate the reviewer's suggestions on the applications. We have conducted additional experiments on applying our SWCNT/BBE as conductive and ultrasoft sensor on the tobacco hornworm. The hornworm body has a much lower modulus (37.7 kPa³³) than those of materials or skin used in our previous demonstrations. We hypothesize that the SWCNT/BBE could be used as a sensor to monitor the physical movements of soft-bodied animals due to its matched softness and satisfied adhesion. For this application, one requirement is that the SWCNT/BBE should have no obvious physical constraints to the hornworm. Thus, we attached our SWCNT/BBE on the back of a hornworm, and recorded its crawling movements. As a comparison, the movement of the same hornworm without attaching the SWCNT/BBE was also recorded. We used the crawling speed (defined as: distance of a single crawling/time) and normalized change of the body length (defined as: body length after each crawling L /initial body length before a crawling L_0) to study the effect of attaching the SWCNT/BBE on the hornworm.

The results show that both the crawling speed and body length change are not affected by attaching the sensor, indicating that our SWCNT/BBE has no obvious mechanical constraints to the physical movements of the hornworm (**Fig. R2-1a-c**). In comparison, a piece of the PDMS Sylgard 184 (with the same dimensions as the SWCNT/BBE sensor) will easily detach from the hornworm body due to high stiffness and limited adhesion (**Fig. R2-1d**). Then, we recorded the resistance change of the SWCNT/BBE sensor with the crawling of the hornworm, and the long-term electrical response shows that our

SWCNT/BBE sensor can easily record the movements of the hornworm without detaching (**Fig. R2-1e**). From this demonstration, it is more straightforward to show the ultra-softness and adhesive properties of the conductive SWCNT/BBE. We believe that our SWCNT/BBE can be used as a promising tool to study the dynamics of crawling for soft-bodied animals and inspire biomimetic design for soft robots.

To make it clear, we have supplied it in the revised main text (page 12 and **Fig. 4**) and supplementary files (**Supplementary Fig. 27, Supplementary Movie 6 and 7**).

Fig. R2-1 a Photographs of the hornworm without attached with the SWCNT/BBE sensor and the one attached with the SWCNT/BBE sensor. **b** The crawling speed and **c** normalized length change of the hornworm. Attaching the SWCNT/BBE sensor has few mechanical constraints to the hornworm compared to that without attached with the sensor. **d** Photographs of the hornworm attached with the PDMS Sylgard 184. The PDMS Sylgard 184 can be easily detached with the hornworm during its crawling due to high stiffness and low adhesion. **e** The resistance change of the SWCNT/BBE sensor with the crawling of the hornworm. The inset shows photographs of the hornworm at the two different states.

For other comments, the point-to-point responses are presented below.

Other technical issues:

Fig. 2b: The strain rate selection is unclear, which questions the stiffness characterization procedure. The Young's modulus should be measured at the

rubber elastic plateau, which should be verified by modulus measurements as a function of frequency. Note that samples with different crosslink densities demonstrate different frequency profiles. Also, the strain rate should be reported in s^{-1} (not in absolute units).

Response:

We thank the reviewer for the valuable comments. We have changed all the units of strain rates to s^{-1} . We have also added rheology measurements (modulus as a function of frequency) of pure BBEs (**Fig. R2-2**). The dynamic frequency sweeps show a nearly linear elastic regime from 0.01 rad/s to 1 rad/s. At lower crosslinking levels (e.g., 1200:1), there is a non-negligible modulus increase at higher frequency (>1 rad/s), possibly due to the low crosslinking level that induces relaxations of polymers⁶. However, all the strain rates (25 mm/min: $0.035 s^{-1}$, 50 mm/min: $0.070 s^{-1}$, 100 mm/min: $0.14 s^{-1}$) used in this work are within the elastic regime (0.01 to 1 rad/s: 0.0016 to $0.16 s^{-1}$).

To make it clear, we have added a sentence in the main text on page 5: “The dynamic sweeps further indicate the elastic regime (0.01 rad/s to 1 rad/s) of the BBE [Supplementary Fig. 6(o)].” We also added the figure and discussions to the supplementary files [Supplementary Fig. 6(o)]:

“The dynamic frequency sweeps show a nearly linear elastic regime from 0.01 rad/s to 1 rad/s. At lower crosslinking levels (e.g., 1200:1), there’s a nonnegligible modulus increase at higher frequency (>1 rad/s), possibly due to the low crosslinking level that induces relaxations of polymer. However, all the strain rates (25 mm/min: $0.035 s^{-1}$, 50 mm/min: $0.070 s^{-1}$, 100 mm/min: $0.14 s^{-1}$) used in this work are within the elastic regime (0.01 to 1 rad/s: 0.0016 to $0.16 s^{-1}$).”

Fig. R2-2 Storage modulus as a function of sweeping frequency of pure BBEs with different crosslinking ratios.

Line 425-426: With regard to the stiffness data in Figures 2 and 3, more details about the fitting model should be provided along with the used fitting parameters and their values.

Response:

Thank you very much for the suggestion. We used the fitting model for unentangled polymer networks based on the literature ⁶:

$$\sigma_{true} = \frac{E}{9} [(\varepsilon + 1)^2 - (\varepsilon + 1)^{-1}] \left\{ 1 + 2 \left[1 - \frac{\beta [(\varepsilon + 1)^2 - (\varepsilon + 1)^{-1}]}{3} \right] \right\}$$

where β is the strand-extension ratio and E is the structural Young's modulus, and σ_{true} and ε are true stress and strain, respectively. We have added the fitting parameters for samples with different crosslinking ratios (molar ratio of monomer:crosslinker = 600:1, 900:1, and 1200:1) at the strain of 400% (**Table R2-2**). The fitting parameters are updated in the revised supplementary table (**Supplementary Table 2**).

Table R2-2. Fitting parameters of PDMS BBEs with different crosslinking ratios at the strain of 400%.

Molar ratio of monomer:crosslinker	E (kPa)	β	ε
600:1	1.787±0.182	0.034±0.003	400%
900:1	1.078±0.158	0.029±0.003	400%
1200:1	0.789±0.198	0.030±0.002	400%

The fitting parameters (E and β) might differ with the selected strain. From the cycling tensile tests, the elongation-at-break of PDMS BBEs should be higher than 400% while 400% strain is the highest strain we selected to conduct the cycling tensile tests. Thus, this table shows the fitting parameters at the strain of 400%.

On line 165, extremely high conductivity values (2.68, 7.08, and 13.78 S/m) are reported, which contradicts to the corresponding values reported in ref 37 using the same TUBALL SWCNT material. In ref 37, conductivity values of 0.01 S/m and 0.09 S/m were reported for SWCNT concentration of 0.25 and 0.51 wt % . Perhaps this discrepancy can be attributed using an alternative method for measuring conductivity.

Response:

Thank you very much for the comments and we apologize for not describing the experiments and methods in detail. In our measurements of the conductivity, we applied a mixture of Ag flakes and liquid metal EGaIn at the contact interface of the probes and the

material (as shown in **Fig. R2-3a** below; **Supplementary Figure 35** in the manuscript). By using the liquid metal mixture, one can effectively reduce the contact resistance⁸. As a result, we can get the resistance of the material itself rather than a value including the contact resistance. We conducted the conductivity measurements on the 0.4 wt% SECNT/BBE sample, and the results show that the measured conductivity (5-7 S/m) by using the liquid metal would be much higher than that (0.01-0.04 S/m) of not using the liquid metal (**Fig. R2-3b**). In Ref. 37, we didn't find any detailed information on the conductivity measurement. If not using the liquid metal mixture to reduce the contact resistance, our values (0.02-0.04 S/m for 0.4 wt% CNT) are at the same level of the results (0.01 S/m for 0.25 wt% CNT, and 0.09 S/m for 0.51 wt% CNT) in Ref. 37. We believe that using the liquid metal mixture is a more accurate way to determine the materials' conductivity properties.

To make it clear, we have further clarified our measuring method on page 19:

“A mixture of Ag flakes and liquid metal EGeIn (10 wt% Ag flakes and 90 wt% EGeIn) was used at the probe-elastomer interface to effectively reduce the contact resistance (**Supplementary Fig. 35**).”

Compared with Ref. 37, our materials have a much lower modulus. In Ref. 37, the moduli of samples with 0 wt%, 0.25 wt%, and 0.51 wt% CNTs are 31 kPa, 66 kPa, and 140 kPa, respectively. While the moduli of our PDMS BBE samples with 0, 0.2, 0.4, and 0.6 wt% CNTs are 0.63 kPa, 2.98 kPa, 5.29 kPa, and 10.65 kPa, respectively. The modulus difference could be attributed to our different chemical components of the bottlebrushes and different crosslinking levels. As our priority was to keep the softness of the conductive BBEs in the soft tissue range (0-30 kPa), we selected a low crosslinking level (crosslinking ratio of MM:CL = 1200:1) to ensure the ultralow Young's modulus (0.63 kPa) of pure PDMS BBE. Thanks to this ultrasoft material matrix, we were able to make the modulus range of SWCNT/BBEs fit within the range of soft biological tissues, and even the highest Young's modulus (10.65 kPa) of our SWCNT/BBE is much lower than that of Ref. 37.

To make it clear, we have added discussions on page 9:

“This could be attributed to our different chemical components of the bottlebrushes and looser crosslinking networks.”

Fig. R2-3 **a** Photographs of the 0.4 wt% SWCNT/PDMS BBE without and with liquid metal as the probe-material interface. **b** Measured conductivity of 0.4 wt% SWCNT/PDMS BBE with and without using the liquid metal.

Line 133-141: I do not agree with the claim to “strong adhesion” properties without at least meeting the shear strength of commonly used adhesive materials. Must be some benchmark to qualify for strong adhesion.

Response:

We truly appreciate the reviewer’s constructive comments. We are sorry to use this improper description for the adhesion property. In the manuscript we have compared our adhesion property with those in literature³⁴ and agree with the reviewer that our shear strength is not as high as those of commonly used adhesive materials. Thus, in the revised manuscript we discuss this difference is possibly due to the inherent lower modulus of our materials and lack of covalent bonds as common adhesive materials do (on page 6):

“Although this level of shear strength is lower than those (>10 kPa) of commonly used adhesive materials³⁴, it is sufficient for securely attaching a piece of BBE to different surfaces in dry conditions, which can withstand its own gravity and different strain levels during operation and even lift objects such as a ping-pong ball or a beaker (potentially useful as a pick-place mechanism; see **Supplementary Fig. 9** and **Supplementary Movie 1**).”

Note that the requirement for the adhesion level can depend on the application scenarios. To this end, we used our materials as adhesives to lift light objects to show that the BBEs have a certain level of adhesive property, and this property could benefit some applications such as wearable electronics. We thank the reviewer again for correcting

this. To make it clear, we have corrected the claim of “strong adhesion” and changed them into “satisfactory adhesion” or “adhesion property” in the manuscript.

Line 222-224: The sentence “Covalent bonds are formed between the SWCNT/BBE 223 and pure BBE during the crosslinking process of the pure BBE insulator thanks to their shared monomers, 224 crosslinkers and thermal initiators,” seems out of place to me.

Response:

Thank you very much for the comment. We apologize that we didn't give detailed experimental results on this part. We have added experiments on showing the bonding of the two types of materials prepared by this crosslinking process. We first prepared a SWCNT/BBE sample by thermal curing, and then added the pure BBE precursor onto the cured SWCNT/BBE sample (the covering area was 13.8 mm length x 10 mm width). After that, we put the SWCNT/BBE and the covering pure BBE precursor together on a hot plate until the top pure BBE precursor was cured into elastomer. With this process, we can get samples of stacked SWCNT/BBE and pure PDMS BBE for the lap-shear test.

As a comparison, we prepared SWCNT/BBE and pure PDMS BBE separately, and then physically attach them with a covering area of 13.8 mm long and 10 mm wide. By conducting the lap-shear test on the two groups of samples, one can find that by curing the SWCNT/PDMS BBE and pure PDMS BBE together, the shear strength and the stretching displacement are higher than those of physically stacked samples (**Fig. R2-4; Supplementary Fig. 21** in the revised Supplementary File). The photographs of the lap-shear test also show that the two pieces of materials can separate easily due to limited bonding. However, by curing the two pieces of materials together, the covering part shows no layer separation (**Fig. R2-4**). The breaking part was on the pure PDMS BBE rather than the bonding part or SWCNT/PDMS BBE, due to the lower Young's modulus of pure BBE. We also supplied a video (**Supplementary Movie 3**) showing the bonding of the SWCNT/PDMS BBE and pure BBE upon stretching. All these results indicate that there is bonding between the two pieces of materials if they are cured together. The same polarity and hydrophilicity of the substrate SWCNT/PDMS BBE and the top BBE precursor allow the diffusion of monomers, crosslinkers, and initiators of the BBE precursor into the SWCNT/PDMS BBE layer. Upon crosslinking, the two layers would form an interpenetrating polymer network at the interface and the two layers were bonded³⁵. To make it clear, we have added discussions on page 9 and supplied the results in the revised supplementary file:

“The monomers and crosslinkers could diffuse into the cured SWCNT/BBE and form an interpenetrating polymer network at the interface between two BBEs upon crosslinking³⁵. As a result, the samples prepared by this curing method show a higher shear strength and better stretchability compared to those of samples prepared by physically attaching (**Supplementary Fig. 21 and Supplementary Movie 3**).”

Fig. R2-4 a The shear strength versus stretching displacement measured between the SWCNT/BBE and pure BBE. The bonding was formed by sequentially curing the SWCNT/BBE and pure BBE, while physically attaching two pieces of samples cannot form bonding at the interface. The inset photographs show the lap-shear test of the two cases. **b** Photographs of the bonded SWCNT/BBE and pure BBE. No detaching was found upon stretching.

Lines 321-324: In addition to all the lightweight materials they use as a proof point for how sensitive the pressure sensors are, it would be good to determine what the minimum mass of detection is for the system.

Response:

Thanks very much for the suggestion. We have added calibration of the sensing strips at the low force range from 0 to 0.01 N (**Fig. R2-5; Supplementary Fig. 30** in the revised Supplementary File). The experimental results show that we can achieve a force sensing value of 0.001 N, at which the normalized resistance changes of sensing strips at the top and bottom, respectively, are 2.26% and 0.19%. The limit of detection was obtained by three sigma limits statistical calculation. As shown in Fig. R2-5, the mean forces and standard deviations are: 0.000157 ± 0.000044 N (top sensing strip), and 0.000154 ± 0.000024 N. Based on the three sigma limits statistical calculation, the limit of detection for the top sensing strip is: $0.000157 + 3 \times 0.000044 = 0.00029$ N, and the limit of detection for the bottom sensing strip is: $0.000154 + 3 \times 0.000024 = 0.00023$ N. Thus, the limit of detection for the touch pad is determined to be 0.00029 N.

Fig. R2-5. **a** Calibration of typical sensing strips at the force range from 0 to 0.1 N, with normalized resistance change of sensing strips as a function of applied force. **b** Measured forces for the blank groups (when the normalized resistance change is 0).

We thank the reviewer for pointing this out. To make it clear, we have updated the limit of detection in the manuscript on page 14 and supplied the figure in the supplementary files (**Supplementary Fig. 30**):

“Based on three sigma limits statistical calculation (**Supplementary Fig. 30c**), the limit of detection of the touch pad was determined to be 0.00029 N.”

Lines 347-349 & fig 5b: I am not convinced that the plot in figure 5b indicates that the impedance from the 0.6 wt% SWCNT/BBE electrodes are in good agreement with those of the hydrogel electrodes because all the samples have a similar profile.

Response:

Thank you so much for the comment. We meant to show the lower impedance when using 0.6 wt% SWCNT/BBE as electrodes on skin. We also attached the Bode plot and Nyquist plot of the impedance measurements under the linear axis (**Fig. R2-6; Supplementary Fig. 33** in the revised Supplementary File). The results show that the impedance of using 0.6 wt% SWCNT/BBE electrodes is smaller than those of using 0.2 wt% and 0.4 wt% SWCNT/BBE electrodes, indicating the higher conductivity of the 0.6 wt% SWCNT/BBE samples. While for the hydrogels, the mechanism of its conductivity is different from that of the SWCNT/BBE (ionic conductivity versus electronic conductivity). So we compared their overall impedance and we found the impedance of using 0.6 wt% SWCNT/BBE electrodes is at the similar level of using the hydrogel electrodes. We think the SWCNT/BBE could be an alternative choice of the hydrogel for the electrodes use.

We removed the sentence: “The impedance data from the 0.6 wt% SWCNT/BBE electrodes are in good agreement with those from the hydrogel electrodes”

We included the sentences on page 15: “The impedance level of using the 0.6 wt% SWCNT/BBE electrodes is similar to that of using the hydrogel electrodes, indicating that our SWCNT/BBE electrode has comparable electrical performance to the gold-standard hydrogel electrode.”

Fig. R2-6. a Bode plot and b Nyquist plot of skin impedance measurements using SWCNT/BBE, copper sheet, and hydrogels as electrodes on the skin.

Line 338: The 0.2, 0.4, and 0.6 wt % SWCNT/BBE mixtures are considered for biomedical applications. However, on lines 210-214, only the cell viability of 0.4 wt% is reported. If the 0.6 wt% is going to be used for biomedical applications, then it should be included in the biocompatibility testing.

Response:

We thank the reviewer to point this out. We have added the viability assay tests for the 0.2 wt% SWCNT/BBE and 0.6 wt% SWCNT/BBE (**Fig. R2-7**). Compared with pure BBE, 0.2 wt%, and 0.4 wt% SWCNT/BBE, the 0.6 wt% SWCNT/BBE has a considerable reduction of the cell viability. We hypothesize that the commercially purchased SWCNT may contain a certain level of cytotoxicity and can lead to the reduction of the viability for samples with higher concentration of SWCNT. Thus, the 0.2 wt% and 0.4 wt% SWCNT/BBE might be more compatible for implantable biomedical applications. In the future, as these materials are used for biological applications, we will investigate this phenomenon.

We updated the figure in the supplementary file (**Supplementary Fig. 19**) and included the sentences in the revised main text on page 9: “The lower viability of the SWCNT/BBE could be attributed to the addition of the SWCNT to the BBE. We hypothesize that the commercially purchased SWCNT may contain proprietary

compounds that could affect the observed results for cytotoxicity. A future investigation will be conducted to determine the presence of potential compounds existing in the SWCNT/BBE.”

Fig. R2-7. Cell viability of human dermal fibroblast cells exposed to pure BBE and conductive BBE (0.2 wt%, 0.4 wt%, and 0.6 wt% SWCNT) relative to counts in the blank control group.

Line 356-358: I do not understand the correlation of the two ECG’s. The authors don’t discuss any comparison. No proof point, they just show two ECG’s and say they’re correlated.

Response:

Thank you very much for the comment. We apologize for not providing a detailed discussion on this point. We have added the overlaid ECG patterns measured by commercial gel electrodes and SWCNT/BBE electrodes (**Fig. R2-8; Supplementary Fig. 34** in the revised Supplementary File). **Fig. 5c** in the manuscript shows that we can use our SWCNT/BBE as an alternative to the commercial gel electrodes for ECG measurement, as the measured ECG pattern is similar with that of using standard commercial electrodes. In addition, we overlaid the two patterns and compared their difference (**Fig. R2-8**). One can clearly find the P waves, QRS complex, and R waves from both patterns³⁶. The amplitude and wave durations (e.g., PR interval and QRS duration) recorded from the SWCNT/BBE electrodes have few differences compared to those measured by the commercial gel electrodes. These results further indicate that the SWCNT/BBE could possibly be a new choice for physiological measurements.

To make it clear, we removed the sentence “From **Fig. 5c**, one can see that ECG signals from our SWCNT/BBE electrodes and the commercial hydrogel electrodes well correlate to each other, indicating comparable performance of our SWCNT/BBE electrode to the

commercial hydrogel electrode”, and have added discussions in the main text and supplied the figure in the supplementary files (**Supplementary Fig. 34**):

“From **Fig. 5c**, one can see that ECG signals can be successfully measured by our SWCNT/BBE electrodes, and show a similar pattern with the one measured by the commercial hydrogel electrodes. In addition, we overlaid the two patterns (**Supplementary Fig. 34**) and one can observe the P waves, QRS complex, and R waves from both patterns ³⁶. The amplitude and wave durations (e.g., PR interval and QRS duration) recorded from the SWCNT/BBE electrodes have few differences compared to those measured by the commercial gel electrodes. These results further indicate that the SWCNT/BBE could be a candidate for physiological measurements.”

Fig. R2-8. The overlaid ECG signal patterns measured by commercial gel electrodes and SWCNT/BBE electrodes.

Reviewer #3 (Remarks to the Author):

dear authors:

I did enjoy reading your manuscript. It is useful and informative. The video files provided are useful and give light to the audience about what you did and difference between samples' behavior. What you claimed in the introduction based on your scenario is realized in your work. I guess some microscopic analyses would be useful, but even at the current form your works take credit for publication.

The methodology, results, and evidences provided as supplementary files are all professionally orchestrated. You also almost completely covered the area, but I think some review papers recently published on this field can be used in the introduction to emphasize classifications made on bottle brush polymers in terms of responsiveness, chemistry, etc. It would attract the general readers. What I propose is to better highlight the necessity of your work in the abstract and also in the introduction.

Regarding analyses, compare your works with similar advanced materials and systems available in the literature. Since your work is practically viable, make good use of patents as well. Technically, you can compare your results in that manner to emphasize the worth of what you did. Moreover, I'd expect a mechanistic explanation for the ending part, in a molecular level. Overall, I did not find something wrong to be mentioned, so you can publish your work after such minor corrections.

Response:

Thanks very much for the positive comments. We have made revisions based on the suggestions.

1. We have compared different methods of preparing bottlebrush elastomers and supplied a table in the revised supplementary files (**Supplementary Table 1**):

Table R3-1. Comparison of different methods of preparing bottlebrush elastomers.

Bottlebrush elastomers	Method	Side chains	Crosslinker	Commercially available	Young's modulus	Reference
PDMS BBE	Free radical polymerization	PDMS (MCR-M11)	PDMS (DMS-R18, DMS-R22)	All materials are commercially available	0.84 - 65.48 kPa	6,23-25

PDMS BBE	Hydrosilylation + platinum catalyst	PDMS (MCR-H21)	PDMS (DMS-H25)	All materials are commercially available	7.43 - 132 kPa	26,27
		PDMS (MCR-V21, MCR-V25)	PDMS-based crosslinker		1.2 - 7.4 kPa	7
linear-bottlebrush-linear elastomers	Atom transfer radical polymerization (ATRP)	PDMS, PMMA	Aggregation of linear polymer tails	No	2.1 - 155.7 kPa	6,28-30
PDMS-r-PEG elastomers	ATRP	PDMS, PEG	Isocyanate:hydroxyl (NCO:OH) and isocyanate:amine (NCO:NH ₂) coupling	No	1.8 - 27.8 kPa	22
ROMP PDMS BBE	Ring-opening metathesis polymerization (ROMP)	PDMS	bis-benzophenone-based PDMS photocrosslinker	No	6.2 - 92 kPa	31
Poly(4-methylcaprolactone) (P4MCL) BBE	Ring-opening metathesis polymerization (ROMP)	P4MCL	4,4'-bioxepane-7,7'-dione	No	10 - 100 kPa	21
PDMS-stat-PEO BBE	Ring-opening metathesis polymerization (ROMP)	PDMS, PEO	bis-benzophenone-based PDMS photocrosslinker	No	32 kPa and 7.7 kPa	19

From the table we can find that the method in this paper is one of the simplest as no complicated chemical synthesis is involved and all the materials are commercially available. The Young's modulus can be ultralow. This could help us to modify the material with new functions and apply them more easily.

2. We have added more discussions in the abstract and introduction to highlight the necessity of this work.

In the abstract, we added:

“To date, an ultrasoft (i.e. Young’s modulus < 30 kPa), conductive, and solvent-free elastomer does not exist.”

In the introduction, we added:

“With the increasing demand of an ultrasoft and conductive material platform to be employed in robotic and biomedical applications, our SWCNT/BBE provides a new solution to build ultrasoft electronics, and will potentially expand our capabilities of understanding biological systems and mimicking their functions.”

3. We have discussed the mechanism of bottlebrush elastomers in a molecular level in the introduction part (on page 3):

“bottlebrush elastomers (BBEs) are a class of intrinsically stretchable materials that can achieve ultra-low Young’s modulus without solvents, because of their highly branched architecture consisting of polymeric side chains attached to a polymer backbone, leading to reduced entanglements in comparison to linear analogues²⁸⁻³³.”

Reference:

1. Lee, S. *et al.* Nanomesh pressure sensor for monitoring finger manipulation without sensory interference. *Science* (80-.). **370**, 966–970 (2020).
2. Wang, Y. *et al.* Robust, self-adhesive, reinforced polymeric nanofilms enabling gas-permeable dry electrodes for long-term application. **118**, 1–10 (2021).
3. Wang, Y. *et al.* A durable nanomesh on-skin strain gauge for natural skin motion monitoring with minimum mechanical constraints. *Sci. Adv.* **6**, 1–10 (2020).
4. Joon, J. *et al.* Antimicrobial second skin using copper nanomesh. 1–11 (2022) doi:10.1073/pnas.2200830119/-/DCSupplemental.Published.
5. Müllner, M. & Müller, A. H. E. Cylindrical polymer brushes – Anisotropic building blocks, unimolecular templates and particulate nanocarriers. *Polymer (Guildf)*. (2016) doi:10.1016/j.polymer.2016.03.076.
6. Vatankhah-Varnosfaderani, M. *et al.* Mimicking biological stress–strain behaviour with synthetic elastomers. *Nature* **549**, 497–501 (2017).
7. Hu, P., Madsen, J. & Skov, A. L. One reaction to make highly stretchable or extremely soft silicone elastomers from easily available materials. 0–34 (2021) doi:10.1038/s41467-022-28015-2.
8. Ohm, Y. *et al.* An electrically conductive silver–polyacrylamide–alginate hydrogel composite for soft electronics. *Nat. Electron.* **4**, 185–192 (2021).
9. Feig, V. R., Tran, H., Lee, M. & Bao, Z. Mechanically tunable conductive interpenetrating network hydrogels that mimic the elastic moduli of biological tissue. *Nat. Commun.* **9**, 1–9 (2018).
10. Ren, X. *et al.* Highly Conductive PPy-PEDOT:PSS Hybrid Hydrogel with Superior Biocompatibility for Bioelectronics Application. *ACS Appl. Mater. Interfaces* (2021) doi:10.1021/acsami.1c04432.
11. Yang, J., Choe, G., Yang, S., Jo, H. & Lee, J. Y. Polypyrrole-incorporated conductive hyaluronic acid hydrogels. *Biomater. Res.* **20**, 1–7 (2016).
12. Tondera, C. *et al.* Highly Conductive, Stretchable, and Cell-Adhesive Hydrogel by Nanoclay Doping. *Small* **15**, (2019).
13. Xia, S., Song, S., Jia, F. & Gao, G. A flexible, adhesive and self-healable hydrogel-based wearable strain sensor for human motion and physiological signal monitoring. *J. Mater. Chem. B* **7**, 4638–4648 (2019).
14. Tringides, C. M. *et al.* Viscoelastic surface electrode arrays to interface with viscoelastic tissues. *Nat. Nanotechnol.* (2021) doi:10.1038/s41565-021-00926-z.
15. Yiming, B. *et al.* Ambiently and Mechanically Stable Ionogels for Soft Ionotronics. *Adv. Funct. Mater.* **31**, 1–11 (2021).
16. Cao, Z., Liu, H. & Jiang, L. Transparent, mechanically robust, and ultrastable

- ionogels enabled by hydrogen bonding between elastomers and ionic liquids. *Mater. Horizons* **7**, 912–918 (2020).
17. Karimkhani, V. *et al.* Tissue-Mimetic Dielectric Actuators: Free-Standing, Stable, and Solvent-Free. *ACS Appl. Polym. Mater.* **2**, 1741–1745 (2020).
 18. Patel, B. B. *et al.* Tunable structural color of bottlebrush block copolymers through direct-write 3D printing from solution. *Sci. Adv.* **6**, 1–14 (2020).
 19. Xie, R. *et al.* Room temperature 3D printing of super-soft and solvent-free elastomers. *Sci. Adv.* **6**, 1–11 (2020).
 20. Mukherjee, S. *et al.* Universal Approach to Photo-Crosslink Bottlebrush Polymers. *Macromolecules* **53**, 1090–1097 (2020).
 21. Self, J. L. *et al.* Dynamic Bottlebrush Polymer Networks: Self-Healing in Super-Soft Materials. *J. Am. Chem. Soc.* **142**, 7567–7573 (2020).
 22. Dashtimoghadam, E. *et al.* Injectable non-leaching tissue-mimetic bottlebrush elastomers as an advanced platform for reconstructive surgery. *Nat. Commun.* **12**, 1–11 (2021).
 23. Duncan, T. T., Chan, E. P. & Beers, K. L. Maximizing Contact of Supersoft Bottlebrush Networks with Rough Surfaces to Promote Particulate Removal. *ACS Appl. Mater. Interfaces* **11**, 45310–45318 (2019).
 24. Ina, M. *et al.* From adhesion to wetting: Contact mechanics at the surfaces of super-soft brush-like elastomers. *ACS Macro Lett.* **6**, 854–858 (2017).
 25. Jacobs, M. *et al.* Nonlinear Elasticity and Swelling of Comb and Bottlebrush Networks. *Macromolecules* **52**, 5095–5101 (2019).
 26. Cai, L. H. *et al.* Soft Poly(dimethylsiloxane) Elastomers from Architecture-Driven Entanglement Free Design. *Adv. Mater.* **27**, 5132–5140 (2015).
 27. Cai, L. H. Molecular understanding for large deformations of soft bottlebrush polymer networks. *Soft Matter* (2020) doi:10.1039/d0sm00759e.
 28. Vatankhah-Varnosfaderani, M. *et al.* Chameleon-like elastomers with molecularly encoded strain-adaptive stiffening and coloration. *Science (80-.)*. (2018) doi:10.1126/science.aar5308.
 29. Keith, A. N. *et al.* Bottlebrush Bridge between Soft Gels and Firm Tissues. *ACS Cent. Sci.* **6**, 413–419 (2020).
 30. Keith, A. N. *et al.* Independently Tuning Elastomer Softness and Firmness by Incorporating Side Chain Mixtures into Bottlebrush Network Strands. *Macromolecules* **53**, 9306–9312 (2020).
 31. Reynolds, V. G. *et al.* Super-soft solvent-free bottlebrush elastomers for touch sensing. *Mater. Horizons* **7**, 181–187 (2020).
 32. Self, J. L. *et al.* Carbon Nanotube Composites with Bottlebrush Elastomers for

Compliant Electrodes. *ACS Polym. Au* (2021)
doi:10.1021/acspolymersau.1c00034.

33. Dorfmann, A. L., Woods, W. A. & Trimmer, B. A. Muscle performance in a soft-bodied terrestrial crawler: Constitutive modelling of strain-rate dependency. *J. R. Soc. Interface* **5**, 349–362 (2008).
34. Yuk, H. *et al.* Dry double-sided tape for adhesion of wet tissues and devices. *Nature* (2019) doi:10.1038/s41586-019-1710-5.
35. Inoue, A., Yuk, H., Lu, B. & Zhao, X. Strong adhesion of wet conducting polymers on diverse substrates. *Sci. Adv.* (2020) doi:10.1126/sciadv.aay5394.
36. Bjerregaard, P. & Gussak, I. Naming of the waves in the ECG with a brief account of their genesis. *Circulation* **100**, 1937–1942 (1999).

REVIEWERS' COMMENTS

Reviewer #1 (Remarks to the Author):

The authors properly addressed my comments. I recommend the publication of the current manuscript.

Reviewer #2 (Remarks to the Author):

The novelty is clearly articulated through comparison with existing synthetic methodologies. Demonstration of new applications is commended. I recommend this paper for publication.

Reviewer #3 (Remarks to the Author):

Dear Authors,

The revised manuscript reflects your ability to present and interpret your work. What I needed are thoroughly provided and discussed. The present version takes credit for publication.

Reviewer #1 (Remarks to the Author):

The authors properly addressed my comments. I recommend the publication of the current manuscript.

Response:

Thank you very much for reviewing our work and we are truly grateful for the insightful comments and constructive suggestions throughout the revision process.

Reviewer #2 (Remarks to the Author):

The novelty is clearly articulated through comparison with existing synthetic methodologies. Demonstration of new applications is commended. I recommend this paper for publication.

Response:

We greatly appreciate your review and valuable suggestions. We are happy to be able to improve our work with reference to your comments. Thanks again for all your insights and support.

Reviewer #3 (Remarks to the Author):

Dear Authors,

The revised manuscript reflects your ability to present and interpret your work. What I needed are thoroughly provided and discussed. The present version takes credit for publication.

Response:

Many thanks for all the positive comments and valuable suggestions on our work. Your precious help and useful inputs are highly appreciated.